# SYNCHROMESH: RELIABLE CODE GENERATION FROM PRE-TRAINED LANGUAGE MODELS

**Gabriel Poesia**[*][†]
Stanford University
poesia@stanford.edu

**Oleksandr Polozov**[*][‡]
X, the moonshot factory
polozov@google.com

**Vu Le, Ashish Tiwari, Gustavo Soares, Christopher Meek, Sumit Gulwani**
Microsoft Research, Redmond
{levu,astiwar,gustavo.soares,meek,sumitg}@microsoft.com

## ABSTRACT

Large pre-trained language models have been used to generate code, providing a flexible interface for synthesizing programs from natural language specifications. However, they often violate syntactic and semantic rules of their output language, limiting their practical usability. In this paper, we propose SYNCHROMESH: a framework for substantially improving the reliability of pre-trained models for code generation. SYNCHROMESH comprises two components. First, it retrieves few-shot examples from a training bank using Target Similarity Tuning (TST), a novel method for semantic example selection. TST learns to recognize utterances that describe similar target programs despite differences in surface natural language features. Then, SYNCHROMESH feeds the examples to a pre-trained language model and samples programs using Constrained Semantic Decoding (CSD): a general framework for constraining the output to a set of valid programs in the target language. CSD leverages constraints on partial outputs to sample complete correct programs, and needs neither re-training nor fine-tuning of the language model. We evaluate our methods by synthesizing code from natural language descriptions using GPT-3 and Codex in three real-world languages: SQL queries, Vega-Lite visualizations and SMCalFlow programs. These domains showcase rich constraints that CSD is able to enforce, including syntax, scope, typing rules, and contextual logic. We observe substantial complementary gains from CSD and TST in prediction accuracy and in effectively preventing run-time errors.

## 1 INTRODUCTION

Large language models (LLMs) trained on massive corpora of unsupervised data have been shown to perform a wide range of tasks, including natural language generation, semantic parsing and sentiment analysis (Brown et al., 2020; Devlin et al., 2019; Raffel et al., 2020). This can be achieved without task-specific training, but rather by adapting the model to each task at test-time using textual *prompts*, which can contain examples and natural language descriptions. In many cases, this methodology was shown to provide good performance, reducing the need to annotate large datasets for each task of interest (Brown et al., 2020; Shin et al., 2021).

An important application of LLMs is in synthesizing programs from natural language descriptions (Austin et al., 2021; Chen et al., 2021). But this task is still challenging for LLMs. First, they can commit *conceptual errors*, generating code that misses the intent behind the given description. For example, when asked to reverse an array, the model might generate code that simply swaps the first and last elements. Indeed, users of current natural language-to-code systems report that models often produce code that is unrelated to their query (Xu et al., 2021).

---

[*]Equal contribution.
[†]Work done during an internship at Microsoft with the PROSE team.
[‡]Work done while at Microsoft Research (polozov@microsoft.com).

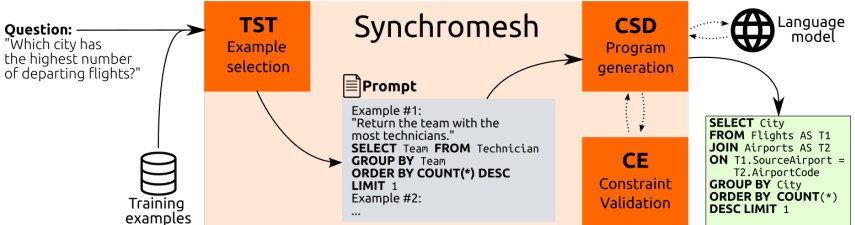

Figure 1: Overview of the SYNCHROMESH framework. Given the user's query, high-relevance examples are first retrieved with Target Similarity Tuning (TST). Then, a program is incrementally sampled via Constrained Semantic Decoding (CSD), which queries a completion engine (CE) to enforce constraints during code generation without re-training or fine-tuning the language model.

Even when they capture the right intent, LLMs can still make *implementation errors*: the generated code can fail to execute. For reversing an array, a model might generate a loop with the correct structure but with an off-by-one error, causing a runtime exception. These errors are common even with very large models. For example, Austin et al. (2021) tested models with up to 137B parameters on generating short Python programs from natural language. Still, 47% of the failures were due to syntax, typing or run-time errors (as opposed to running but producing incorrect output). This is in line with theoretical results in Merrill et al. (2021) showing that programming language semantics cannot be fully inferred from ungrounded data. Together, both observations suggest that simply scaling up LLMs might be ineffective to obtain reliable performance, especially for longer programs.

In this paper, we address both conceptual and implementation errors with SYNCHROMESH, a framework for reliable code generation from pre-trained models. Since LLMs are highly sensitive to which few-shot examples are given in their prompt, we propose Target Similarity Tuning (TST): a method for dynamically selecting semantically relevant examples for a given description. TST mitigates conceptual errors by learning to select examples with similar intent, even when their natural language descriptions seem unrelated in form. Given relevant examples, we then generate programs with Constrained Semantic Decoding (CSD), a novel method for enforcing rich syntactic and semantic constraints during code generation on top of a frozen language model. Rich language-specific constraints, ranging from syntax validity to scoping and type-checking, can be implemented under the simple abstraction of *completion engines (CE)*. CSD aligns these constraints with the language model's token vocabulary by leveraging Brzozowski language derivatives (Brzozowski, 1964). This guarantees that all sampled programs satisfies the implemented constraints, preventing whole classes of implementation errors by construction. The pipeline is illustrated in Figure 1.

We demonstrate the generality of SYNCHROMESH in three real-world languages: SQL (database queries), Vega-Lite (data visualization) and SMCalFlow (calendar applications). In experiments with GPT-3 and Codex, we observe that SYNCHROMESH can eliminate whole classes of errors that make outputs from unconstrained models either fail to execute or produce trivial results (e.g., empty charts). Furthermore, eliminating invalid programs consistently improves prediction accuracy. In summary, we make the following contributions:

- We propose Target Similarity Tuning for selecting few-shot examples based on the similarity of the programs they describe, improving relevance and downstream performance.
- We introduce completion engines as an abstraction that can implement rich classes of semantic program constraints, as we demonstrate in SQL, Vega-Lite and SMCalFlow.
- We introduce a general, constraint-observing decoding algorithm, which aligns programming language constraints with the language model's token vocabulary.
- We evaluate our method in three natural language-to-code tasks. CSD and TST both show strong complementary gains in output validity and prediction accuracy across domains.

## 2 TARGET SIMILARITY TUNING

In this section, we first overview the challenge posed by conceptual errors in programs synthesized by LLMs. We then introduce TST, which improves performance through more relevant example

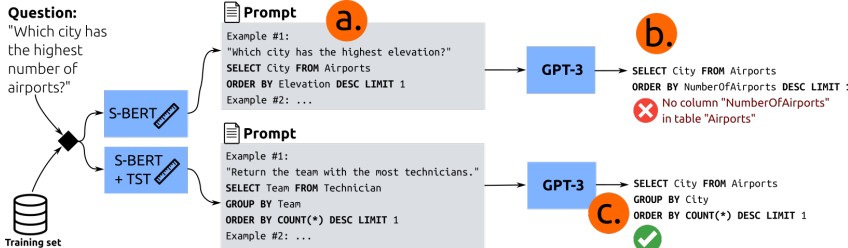

Figure 2: Example of Target Similarity Tuning improving example selection for synthesizing a SQL query. In (a), the prompt example missed the key query structure (grouping and counting). With this example, GPT-3 generates an invalid query (b). With TST, we retrieve a relevant example which GPT-3 successfully adapts to answer the user's question (c).

selection. Throughout, we will use a real example of synthesizing a SQL database query to answer a question posed in natural language.

Suppose a data analyst has a relational database of airports and wants to answer the following question: "Which city has the highest number of airports?" One procedure for turning this description into a SQL query is to use an LLM such as GPT-3 (Brown et al., 2020) or Codex (Chen et al., 2021). To prompt the model for the task at hand, we would feed it with a natural language description of the task and a selection of input-output examples.

Given the analyst's question, how do we select the most relevant examples from a training pool? Liu et al. (2021a) proposed to retrieve examples with *similar natural language descriptions* using a pre-trained paraphrase detection model. Figure 2a shows the most similar example from the Spider natural language-to-SQL dataset (Yu et al., 2018) according to Sentence-BERT (Reimers & Gurevych, 2019). The query "Which city has the highest elevation?" is similar on a surface level: it also asks "Which city has the highest □?". This training query asks about "elevation", a property that is readily available as a column in the Airports table. Figure 2b shows GPT-3's output when given this and a few other examples. The model attempts to mimic the top example, referring to a nonexistent column "NumberOfAirports". The issue is that we picked the example in the prompt based on *description similarity* and not *SQL query similarity*. In fact, the SQL query in the chosen example had a simplistic structure that was significantly different from the structure of the desired SQL query, and this contributed to the failure at Point (b) in Figure 2.

We want to retrieve examples that have relevant program structures for the test query. We do so using our fine-tuning scheme called Target Similarity Tuning (TST). Formally, suppose $\mathcal{D}$ is a dataset of programs and associated utterances, with $\mathcal{D}_i = (p_i, u_i)$. Let $S(p_a, p_b) \in [0, 1]$ denote a normalized similarity metric between programs. If $f_\theta$ is a pre-trained similarity model for natural language sentences, TST consists in fine-tuning $f$ to predict the similarity between *target programs* given by $S$ from their descriptions. Precisely, we minimize the mean-squared error loss:

$$\mathcal{L}_{TST}(\theta) := \mathbb{E}_{i,j \sim \mathcal{D}} \left[ f_\theta(u_i, u_j) - S(p_i, p_j) \right]^2 \ .$$

We define $S$ using the classical tree edit distance algorithm from Zhang & Shasha (1989) to compare Abstract Syntax Trees (ASTs). Figure 2c shows GPT-3's output when given examples selected with TST. Now, the output query is correct: it performs a "group by" on the "City" column, and sorts by the count of records in each group. This structure was already present in the top example selected by TST, corresponding to "Return the team with the most technicians". Even if the analyst's question and this utterance are drastically different in natural language, they share similarity in the SQL query that they describe. The TST objective is able to properly capture this fact. As our experiments show in Section 4, TST significantly boosts the performance of both GPT-3 and Codex.

## 3    CONSTRAINED SEMANTIC DECODING

We now present Constrained Semantic Decoding (CSD) as an approach to eliminate implementation errors from code generated by LLMs. We first illustrate CSD with an example, and then formalize it using the abstraction of CEs.

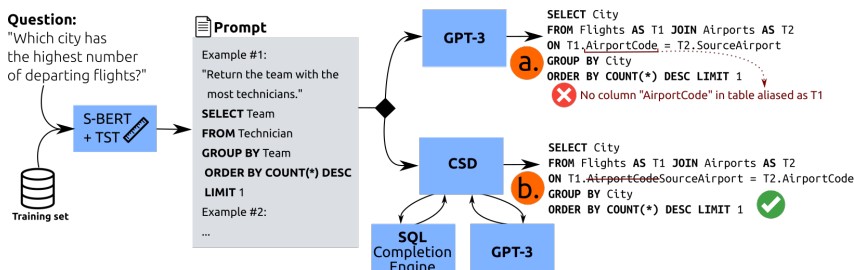

Figure 3: Example on CSD generating a SQL query. Given the prompt, GPT-3 makes a mistake (a) when generating the `JOIN` condition. CSD is able to prevent this error by (b) keeping track of table aliases and constraining the model to respect the database schema.

The example in Figure 2 showed that TST can help LLMs generate the correct program. In general, however, TST only helps LLMs by guiding toward the correct structure, but the model still needs to fill all the specific implementation details correctly. Figure 3 shows a case where the model cannot simply adapt one example from the prompt. Here, the user's query is "Which city has the highest number of departing flights?" This query is similar to the previous one – in fact, TST retrieves the same top-1 example as before. But now the correct SQL query needs to join the "Airports" and "Flights" tables. GPT-3 generates the join condition $Flights.AirportCode = Airports.SourceAirport$, but this condition has a subtle error: the column names of the two tables are swapped. Thus, this query fails to execute. In general, unconstrained language models often make such implementation errors: using undeclared variables, losing track of nesting levels when producing complex expressions, or calling functions using arguments of the wrong type. Even the smallest of such errors prevents generated code from executing.

CSD prevents implementation errors *by construction* (as opposed to repairing after-the-fact). Imagine we have access to an oracle, which we call a *CE*, that can take a partial program and return all tokens that can extend that partial program toward a complete correct program. When the LLM is generating the program token by token, CSD ensures that the next token is sampled from the set returned by the CE.

In Figure 3, after generating "`T1.`" inside the "on" clause, our SQL CE resolves the alias and constrains the model to output one of the columns from the "Flights" table. This fixes the error seen previously during generation and produces the correct SQL query.

### 3.1 COMPLETION ENGINES

We now formally define CEs. Let $\Sigma$ be a base alphabet, and $\Sigma_L \subseteq \Sigma^*$ be the (potentially infinite) set of tokens of the target language. Our goal is to sample programs from a language $L \subseteq \Sigma_L^*$ – the set of valid programs. A CE $C_L$ is a partial function from $\Sigma_L^*$ to a set of tokens. We use a regular expression over $\Sigma$ to represent a set of tokens. The strings in the domain of $C_L$ are called *completion points*, and a CE satisfies the following axioms: (A1) The empty string and every $p \in L$ must be completion points. For every $p \in L$, $C_L(p) = r'\$'$, where $r'\$'$ is the regular expression that matches the stop token. (A2) If $s \in \Sigma_L^*$ is a completion point and $t$ fully matches $C_L(s)$, then their concatenation $st$ must also be a completion point. (A3) The CE is *exhaustive*; that is, if $s$ is a completion point and $s = tt_0$, where $t_0$ is a token, then $t$ should be a completion point and $C_L(t)$ should match $t_0$. Furthermore, we assume that CEs are only called after maximal matches. For example, if a partial program ends in an identifier, the CE can assume that the identifier is complete.

Our CEs are implemented in two layers: a context-free layer, which enforces syntactic validity, and a context-sensitive layer, which encodes semantic constraints that depend on language semantics and *the user's context* (e.g., the database). Below, we describe an automatic method for constructing context-free CEs directly from the target language's grammar. The context-sensitive layer of an engine is specific to the target language. Table 1 provides an overview of several constraints implemented by our CEs for SQL, Vega-Lite and SMCalFlow, three rich languages with different syntactic and semantic structures. A detailed description of the three CEs can be found in Appendix C.

| Language | Constraint | Example of partial program | Valid/Invalid Examples |
|---|---|---|---|
| SQL | A valid identifier must follow after `AS`. | `SELECT` Name, Role `FROM` User `AS` ∧ | U ✓
T1 ✓
2 ✗ |
| | Column names must come from schema, even behind aliases. | `SELECT` U.Name `FROM` User `AS` U `WHERE` U. ∧ | Name ✓
DoB ✓
Birthday ✗ |
| Vega-Lite | Data fields must be used with compatible types. | `{"x": {"field": "Category", "type":` ∧ | `"nominal"` ✓
`"temporal"` ✗ |
| | Do not facet on field with too many distinct values (breaks rendering). | `{"column":{"field":` ∧ | `"Category"` ✓
`"ZipCode"` ✗ |
| SMCalFlow | Type-check parameters of all API functions. | `(Yield (PlaceHasFeature (` ∧ | `Takeout` ✓
`IsWindy` ✗
`List.Apply` ✗ |
| | Track declared variables and their types. | `(let (x 85) (Yield (inCelsius` ∧ | `x` ✓
`y` ✗ |

Table 1: Examples of constraints implemented in our CEs for SQL, Vega-Lite and SMCalFlow. Given a partial program, CEs return a regular expression that matches the valid tokens that can follow. Here, we show positive and negative token examples for each such regular expression. This abstraction allows domain experts to encode a wide range of expressive code generation constraints.

**Deriving completions from grammars**  Computer language parsers are often automatically generated from a grammar. The grammar contains enough information to derive the context-free layer of CEs. To facilitate this process, we created a library that extends any parser generated by ANTLR (Parr & Fisher, 2011), a popular LL(*) top-down parser generator, to provide token-level completions. Namely, we **(i)** let the ANTLR-generated parser process the given program prefix $p$, **(ii)** retrieve its state in the Augmented Transition Network (ATN) at the last program token, **(iii)** traverse the ATN from that state to enumerate all possible next token productions. This process yields **(a)** a list of productions and token types $\{\tau_j\}_{j=1}^K$ that are allowed to follow $p$ and **(b)** a partial AST $T_p$. Each CE takes $\{\tau_j\}$ and $T_p$ as input to generate semantic context-sensitive constraints.

## 3.2 FROM COMPLETION ENGINES TO A DECISION PROCEDURE

We use CEs to guide sampling from an LLM. A key component of our algorithm for constrained sampling is a decision procedure for membership in prefix-closure of the set $L$ of all valid programs. The prefix-closure $L^c$ of a language $L$ contains all programs in $L$ as well as all of their prefixes. Intuitively, $L^c$ contains all partial programs that can be *completed* to a valid program. Given a CE $C_L$, our first goal is to build a decision procedure for $L^c$: given a string $s$, does it belong to $L^c$?

We answer if $s \in L^c$ by repeatedly calling $C_L$ on certain prefixes $p$ of $s$ and matching the regular expression $C_L(p)$ with suffixes of $s$. We start with $p$ being the empty string. We find the maximal prefix of $s$ that matches the regular expression $C_L(p)$ and remove it from $s$ and add it to $p$, and repeat until the match fails. There are two cases: either $s$ is empty now, which means the input string was a completion point and hence it is in $L^c$, or $s$ now is the *remainder* left after removing the largest prefix that was a completion point. For the second case, we must check: does there exist a completion string $c$ such that $sc$ fully matches the regular expression $C_L(p)$?

This question can be efficiently answered by *Brzozowski derivatives* (Brzozowski, 1964). Formally, the derivative of a formal language $S$ with respect to a string $u$ is another formal language $u^{-1}S = \{v : uv \in S\}$. In other words, it is precisely the set of strings that can complete $u$ to some string in $S$. If $u^{-1}S = \emptyset$, then no string in $S$ starts with $u$. Brzozowski derivatives are efficient to compute for our regular languages (or regular expressions defining them) – we describe a simple linear-time algorithm in the Appendix. Given the derivative of $C_L(p)$, answering whether $s$ can be completed to

belong to $C_L(p)$ reduces to performing a simple regular expression match. This operation answers the case when the remainder is non-empty and completes our decision procedure for $L^c$.

## 3.3 THE CONSTRAINED SEMANTIC DECODING ALGORITHM

Using the decision procedure for $L^c$, we can now describe the Constrained Semantic Decoding algorithm. Suppose $s \in L^c$ is the language model's output so far (we start with $\epsilon$). If $\Sigma_M$ is the model's vocabulary, we can compute the set of valid next tokens $V_M(s) = \{t \in \Sigma_M : st \in L^c\}$ by using our decision procedure for each token in the vocabulary $\Sigma_M$. In other words, we maintain the invariant that the model's current partial output $s$ is in $L^c$, and make progress by using the model to sample from $V_M(s)$, instead of the unconstrained $\Sigma_M$. Once we have a complete program, we are guaranteed that it will satisfy all constraints enforced by the CE.

One subtlety to note is that language models and programming languages have drastically different tokenizations; i.e., $C_L$ and LLM work with different tokens. For instance, a long string literal is a single SQL token, but might span multiple tokens for the language model. Similarly, a single token from the language model's vocabulary might close multiple parentheses at once. In general, token boundaries between the two can be arbitrarily misaligned. Each decision of whether $st$ belongs to $L^c$ can potentially cross multiple completion points, or might not even finish a maximal match to the previous completion point (see the Appendix for an example prediction in Vega-Lite where this happens multiple times). Nevertheless, our CSD algorithm described here naturally handles this alignment problem. Hence, in SYNCHROMESH, CEs do not need to be aware of this issue – they can be fully implemented in terms of the target language's tokens.[1]

Our implementation applies substantial optimizations that leverage the structure of Byte-Pair Encoding vocabularies (namely, that many tokens are prefixes of longer tokens) and reuse computation. We detail these optimizations in Appendix E. In our experiments with GPT-3, CSD adds an average of 8% overhead to the sampling procedure – a relatively small impact to trade for output correctness.

## 4 EXPERIMENTS

We evaluate SYNCHROMESH in three tasks of synthesizing code from natural language descriptions. For SQL, we use the Spider dataset (Yu et al., 2018). For Vega-Lite, we use the NLV Corpus (Srinivasan et al., 2021). For SMCalFlow, we use the dataset that introduced the language (Andreas et al., 2020). In NLV, which has visualizations over 3 different datasets, we alternate using each dataset as a test-set by only using training examples from the other two datasets. In Spider and SMCalFlow, we use the training/validation set split given in each dataset.

**Example selection model**  To select examples, we use Sentence-BERT (Reimers & Gurevych, 2019) to fetch the 5 closest examples by cosine similarity. When using TST, we fine-tuned the model with the TST objective in both the Spider and SMCalFlow training sets. The NLV corpus is smaller and does not provide a clear train-test split to fairly evaluate TST. Holding out one dataset and fine-tuning on the remaining two yields SYNCHROMESH *accuracies* of over 90%. However, we attribute that performance to the fact that NLV has only 10 distinct visualizations and the same participants labeled all three datasets. For that reason, we omit Vega-Lite from the TST experiments.

**Language models**  We used the two largest models from the GPT-3 family (Brown et al., 2020, with 13B and 175B parameters), as well as the largest Codex model (Chen et al., 2021). Codex shares the same architecture with 175B GPT-3, but its training set contained a larger portion of source code in a variety of languages. Our only access to the models was through the public OpenAI HTTP API, which allowed us to apply constraints by adding a bias to logits. We describe the necessary adaptations of CSD to this setting in Appendix F.

**Metrics**  For Vega-Lite and SMCalFlow, we report the *exact-match accuracy* between predictions and ground-truth (field order is disregarded in Vega-Lite). In SQL, we instead measure *execution*

---

[1]CSD aligns the LLM's stream of vocabulary (sub-)tokens to the CE's stream of valid program completion points, akin to a clutch that dynamically aligns the speeds of differently-sized gears in a manual transmission. Such mechanism is known as *synchromesh* (Jewkes et al., 1969), which gives the name to our whole framework.

| Model | SQL | | | Vega-Lite | | | SMCalFlow | | |
|---|---|---|---|---|---|---|---|---|---|
| | Exec. | Valid | Dist. | Acc. | Valid | Dist. | Acc. | Valid | Dist. |
| Andreas et al. (2020) | - | - | - | - | - | - | 72%[S] | - | - |
| Srinivasan et al. (2021) | - | - | - | 64%[S] | - | - | - | - | - |
| Rubin & Berant (2021) | 71%[S] | - | - | - | - | - | - | - | - |
| Scholak et al. (2021) | 79%[S] | 98% | - | - | - | - | - | - | - |
| GPT-3 13B | 16% | 43% | 0.42 | 14% | 55% | 0.51 | 38% | 76% | 0.43 |
| ” + CSD | 20% | 66% | 0.44 | 17% | 100% | 0.48 | 40% | 95% | 0.40 |
| ” + TST | 14% | 48% | 0.42 | - | - | - | 60% | 88% | 0.22 |
| ” + CSD + TST | 19% | 72% | 0.43 | - | - | - | 63% | 98% | 0.17 |
| GPT-3 175B | 28% | 49% | 0.36 | 20% | 67% | 0.36 | 44% | 77% | 0.41 |
| ” + CSD | 35% | 73% | 0.36 | 25% | 100% | 0.32 | 45% | 97% | 0.37 |
| ” + TST | 31% | 56% | 0.35 | - | - | - | 60% | 88% | 0.24 |
| ” + CSD + TST | 37% | 76% | 0.34 | - | - | - | 66% | 97% | 0.18 |
| Codex 175B | 56% | 73% | 0.25 | 39% | 87% | 0.24 | 45% | 79% | 0.37 |
| ” + CSD | 61% | 85% | 0.23 | 40% | 99% | 0.23 | 46% | 97% | 0.33 |
| ” + TST | 60% | 81% | 0.23 | - | - | - | 63% | 90% | 0.21 |
| ” + CSD + TST | 64% | 85% | 0.23 | - | - | - | 63% | 99% | 0.19 |

Table 2: Results of each language model on all domains with and without CSD and TST. For SQL, we run the resulting query and report Execution Match accuracy (**Exec.**). For Vega-Lite and SM-CalFlow, we instead report Exact Match accuracy (**Acc.**). Edit Distance (**Dist.**) measures average relative edit distance between the prediction and the ground truth. We also report the fraction of **Valid** model outputs (those that parse, type-check and execute). For context only, we show recent results from *supervised* models (trained on the datasets we use) marked with [S].

*accuracy*, comparing query results instead. For a more fine-grained signal, we additionally measure the *edit distance* between the predicted and ground-truth ASTs using the normalized tree edit distance (Zhang & Shasha, 1989).

## 4.1 RESULTS

Table 2 and Figure 4 summarize our main results evaluating SYNCHROMESH. Key observations are:

SYNCHROMESH *improves reliability on top of all pre-trained LLMs.* First, it improves top-1 accuracy (exact or execution-measured) over any pre-trained LLM in all domains. SMCalFlow benefits the most, likely because this domain-specific language is absent in the LLM pre-training corpus. For SQL and SMCalFlow, the absolute gain is almost the same for equally-sized GPT-3 and Codex.

Second, SYNCHROMESH dramatically improves validity. In SQL, it eliminates execution errors from 29% of the queries generated by GPT-3 13B (as validity improves from 43% to 72%). Even Codex benefits, with 12% more queries executing successfully after SYNCHROMESH augmentation. In Vega-Lite and SMCalFlow, SYNCHROMESH improves reliability even more substantially. GPT-3 13B only produces valid charts for 55% of the queries in NLV; all errors are eliminated with SYNCHROMESH. This is nearly paralleled in SMCalFlow, in which all models produce well-typed programs 97% of the time or more with SYNCHROMESH.

SYNCHROMESH *brings the output closer to ground truth.* Error prevention alone is trivial (e.g., with a constant error-free prediction), but not while simultaneously improving accuracy or edit distance to the ground-truth, as SYNCHROMESH does. Again, we observe improvements in all domains and the most in SMCalFlow. For GPT-3 175B, the average edit distance is reduced from 0.41 to 0.18.

*TST and CSD bring complementary benefits.* Our ablation studies reported in Table 2 show that their combination performs better than either one separately. TST helps LLMs generate programs in the "vicinity" of the correct one, and CSD helps by "guiding" the models toward the correct one.

SYNCHROMESH *adds more value for longer programs.* Program synthesis is hardest when the target program is complex. Does SYNCHROMESH improve synthesis of longer programs, or are its benefits

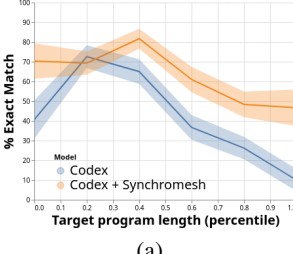 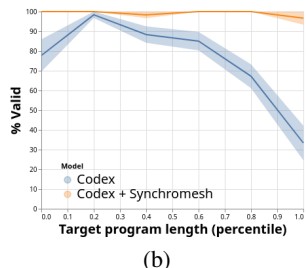 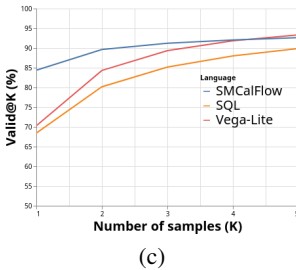

Figure 4: (a) Accuracy and (b) validity of Codex predictions with and without SYNCHROMESH on SMCalFlow as a function of the ground-truth program length. We map program lengths to percentiles, and round to the closest multiple of 10%. Error bands correspond to standard error. (c) Evaluation of the "generate-then-test" approach with Codex, showing the probability of at least one prediction being a valid program (Valid@K) for up to 5 samples.

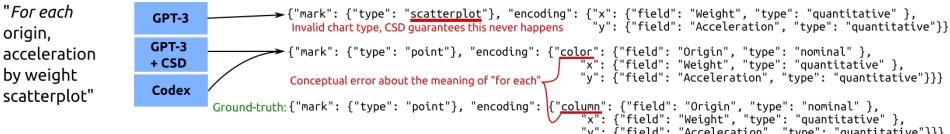

Figure 5: Illustration of implementation and conceptual errors in Vega-Lite. CSD can avoid generating the invalid Vega-Lite mark type "scatterplot", though conceptual errors can still remain.

coming from fixes to small programs? Figure 4(a) shows accuracies and (b) validity for SMCalFlow broken down by the length of the ground truth program (we show results for SQL in the Appendix). Here, program lengths are shown as their percentile. With SYNCHROMESH, we see that accuracy decays at a slower pace, and validity remains high throughout, when compared to Codex alone. This indicates that SYNCHROMESH *improves the ability of base models to generate longer programs.*

*LLMs augmented with* SYNCHROMESH *approach but underperform supervised models.* For context, we include state-of-the-art results at the time of writing for each task in Table 2. We note that these methods fine-tune or train the underlying language-to-code model on each task, thus are not directly comparable to LLMs with SYNCHROMESH. That said, we observe that base LLMs—even Codex— substantially underperform supervised models (19% worse for SQL; 27% worse for SMCalFlow), and SYNCHROMESH helps narrow that gap (now 11% worse for SQL; 9% worse for SMCalFlow).

SYNCHROMESH *outperforms "generate-then-test".* CSD enforces program constraints during generation. Instead, prior work has leveraged a "generate-then-test" approach: take multiple samples and filter out those that produce errors or violate constraints (Chen et al., 2021). Figure 4(b) evaluates this approach with Codex, the highest performing base LLM. We sample from Codex with a temperature $\tau = 0.7$ to obtain diverse but high-quality samples. We then compute the "Valid@K" metric by using the "Pass@K" estimator from Chen et al. (2021) to calculate the probability of at least one valid sample among $K$, with $K \leq 5$. In SQL, Codex needs 3 samples to match SYNCHROMESH (Valid@K = 85%). In SMCalFlow and Vega-Lite, SYNCHROMESH is able to virtually eliminate errors with 1 sample, while "Valid@5" for Codex is still below 93%. This provides evidence that even the best LLMs benefit from incremental validation, especially in less popular languages.

## 4.2 DISCUSSION

In our experiments, SYNCHROMESH was able to improve accuracies and program validity across all languages due to better examples to use as a reference (TST) and preventing errors during generation (CSD). Yet, this approach has an important limitation. While TST can reduce conceptual errors, and CSD can guarantee that certain implementation errors never occur (e.g., type errors in SMCalFlow, or undefined column references in Vega-Lite or SQL), TST cannot guarantee elimination of conceptual errors. When those occur, CSD is usually insufficient to correct the prediction. Figure 5 shows an example in Vega-Lite from the "Cars" dataset in NLV. Here, the user asks for one scatter plot

*for each* origin, indicating faceting (multiple charts). GPT-3 alone produces an invalid Vega-Lite chart type, "scatterplot". CSD can eliminate this error, guiding GPT-3 to generate "point" instead. However, a conceptual error remains: instead of faceting, the model colors points by their origin. Codex produces the correct Vega-Lite mark type, but still makes the same conceptual mistake.

Nonetheless, we argue that improving validity is especially important for user-facing applications. Users of language-to-code systems might need to rephrase their request or to edit the system's output. But outputs that fail to even execute undermine user experience: fixing an automatically generated program can be more cumbersome than writing it in the first place. In LLM-driven systems like Github Copilot, implementation errors can remain unnoticed and introduce bugs or vulnerabilities.

## 5 RELATED WORK

Program synthesis is a long-standing AI challenge with the goal of generating computer programs from higher-level specification (Gulwani et al., 2017). In particular, synthesis from natural language descriptions has gained recent attention (Liu et al., 2016; Yaghmazadeh et al., 2017), thanks to advances in natural language processing models such as Transformers (Vaswani et al., 2017). Typically, LLMs such as GPT-3 (Brown et al., 2020) and Codex (Chen et al., 2021) output an unconstrained sequence of tokens, and still often make conceptual or implementation errors in generated programs (Austin et al., 2021). Specialized training, e.g. to output an AST (Wang et al., 2020; Yin & Neubig, 2017), can mitigate *syntactic* errors, but still does not guarantee accuracy or conformance to domain-specific *semantic* constraints. Moreover, it requires a specialized architecture and a decoding procedure for each target language. Instead, SYNCHROMESH applies such constraints at inference, neither using specialized architectures nor fine-tuning the LLM.

The general idea of constraining LLMs when generating programs has been explored in recent work. Shin et al. (2021) applied syntactic constraints for semantic parsing. However, their method requires enumerating *all* valid programs for determining valid next tokens for the LLM, and does not enforce semantic constraints. In concurrent work, Scholak et al. (2021) applied similar semantic constraints to synthesizing SQL queries. The authors substantially improve the performance of an already fine-tuned model by leveraging an incremental parser. We see CSD as a generalization of these efforts, as our completion engines can apply context-sensitive constraints by dynamically constructing regular expressions. Aligning these constraints with the underlying model vocabulary does not require fine-tuning: SYNCHROMESH only trains the much smaller target similarity model (Section 2).

Since the emergence of LLMs, researchers have developed numerous techniques to adapt them to new domains (Liu et al., 2021b). Many focus on *prompting*, i.e. generating a domain- and instance-specific input to an LLM to increase the likelihood of correctness. In *few-shot prompt augmentation*, Gao et al. (2020) use pre-trained sentence embeddings to select the closest prompt examples to the given input instance. Liu et al. (2021a) further fine-tune sentence embeddings on the available training set of input utterances. TST in SYNCHROMESH takes this approach a step further, and fine-tunes the embedding models based on *output* similarity. It optimizes the amount of relevant output bits in the prompt, thereby reinforcing the necessary hints for the LLM.

## 6 CONCLUSION

SYNCHROMESH augments program synthesis with pre-trained LLMs to prevent conceptual and implementation errors during generation. We designed SYNCHROMESH to be easily usable with minimal NLP or LLM knowledge expected from a domain expert. As such, it **(a)** automatically generates the completion engine API from the language grammar, **(b)** does not require fine-tuning the LLM, drastically reducing the data/compute budget, and **(c)** integrates into the decoding loop or inference API with minimal overhead. Our method significantly improves performance of both GPT-3 and Codex in three languages, both by boosting accuracy and consistently improving output validity.

While real-world and well-established, the domains we study are still not Turing-complete. We envision extending SYNCHROMESH to a Turing-complete language like Python can vastly increase reliability of LLM-based systems like Github Copilot. This requires further extension of CSD to integrate with the parser/interpreter of the target language, and to study applicable classes of constraints. The TST technique, however, can be used in any LLM-based language-to-code system.

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

| **Algorithm 1:** CSD$(M, \Sigma_M)$ | **Algorithm 2:** ValidPrefix$(s)$ |
|---|---|
| **Input** : A LLM-based token generator, $M$ | **Input** : A string $s$ |
| **Input** : Its token set $\Sigma_M$ | **Output:** True iff $s \in L^c$ |
| **Output:** String generated by $M$, constrained by $C_L$ | $p \leftarrow next\_prefix \leftarrow$ ""; |
| $s \leftarrow next\_token \leftarrow$ ""; | **while** $next\_prefix \neq \bot$ **do** |
| **while** $next\_token \neq$ "\$" **do** | $\quad$ $p \leftarrow p \cdot next\_prefix$; |
| $\quad$ $valid\_tokens \leftarrow \{t \in \Sigma_M \mid$ ValidPrefix$(st)\}$; | $\quad$ $s \leftarrow next\_prefix^{-1} \cdot s$; |
| $\quad$ $next\_token \leftarrow$ Sample$(M(s), valid\_tokens)$; | $\quad$ $regex \leftarrow C_L(p)$; |
| $\quad$ $s \leftarrow s \cdot next\_token$; | $\quad$ $next\_prefix \leftarrow$ startswith$(s, regex)$; |
| **end** | **end** |
| **return** $s$; | **return** $(s^{-1} \cdot regex \neq \{\})$; |

Figure 6: ValidPrefix$(s)$ determines if $s$ is in the prefix-closure $L^c$ of $L$ given the completion engine $C_L$ for $L$. CSD$(M, \Sigma_M)$ returns a string generated by the model $M$ ensuring that the string's underlying token sequence is consistent with the completion engine $C_L$.

## A    CONSTRAINED SEMANTIC DECODING ALGORITHM

In Section 3.3, we described the Constrained Semantic Decoding algorithm in the text. We provide the same algorithm in pseudo-code in Algorithms 1 and 2 in Figure 6 below. ValidPrefix is our decision procedure for $L^c$, while CSD samples a complete program from the model $M$ by making calls to ValidPrefix. We use $s \cdot t$ to denote concatenation of $s$ and $t$, and $s^{-1} \cdot t$ to denote the string obtained by removing the prefix $s$ from $t$. The utility function startswith$(s, r)$ returns the maximal prefix of $s$ that matches the regular expression $r$, and returns $\bot$ if there is no such match. The function Sample$(Dist, S)$ returns a token from the set $S$ of tokens sampled from the distribution $Dist$ restricted to $S$. The CSD procedure uses the model $M$ on the partial program to generate a distribution on the next token, but constrains it to belong to the set of $valid\_tokens$ determined by the completion engine $C_L$.

## B    COMPUTING BRZOZOWSKI DERIVATIVES

Regular expression derivatives can be expensive to compute in general. The main challenge is that Klenee stars fork the decision procedure: the algorithm must decide to repeat or skip the pattern inside a star. However, in this work, all our regular expressions come from grammars of programming languages. These languages have one important feature in common: tokens are defined by greedy matching rules. This means that Klenee stars consume as many characters as possible, and do not backtrack if a tokenization error occurs (the error simply propagates). Under the assumption that Klenee stars have greedy semantics, derivatives can be computed in linear time. The algorithm for computing derivatives of a regular expression can be worked out by looking at all constructors of regular expressions. Table 3 details this computation for regular expressions of a base alphabet $\Sigma$.

## C    COMPLETION ENGINES

Here, we describe our completion engines for SQL, Vega-Lite and SMCalFlow in more detail.

### C.1    SQL

SQL database queries are executed in the context of a particular database containing a set of tables. Each table has a schema, which specifies named columns, their data types and constraints such as foreign keys. We refer to Yu et al. (2018) for a more detailed description of the SQL language.

Our CE for SQL enforces that only columns that exist in the tables in the database are used. The main challenge is that queries often specify aliases. Thus, during parsing, we construct a symbol table mapping aliases to the tables they refer to. We enforce that tables that already have an alias should only be referred to by their alias, not in their unqualified form. Since aliases can be referred to in the SELECT clause before being defined in the FROM clause, we also keep track of undefined

| Constructor | Description | Derivative w.r.t $c' \in \Sigma$ |
|---|---|---|
| $\varnothing$ | Empty regular expression (matches no string). | $\varnothing$ |
| $\epsilon$ | Matches only the empty string. | $\varnothing$ |
| $c$ | Matches a single character $c$ | $\epsilon$ if $c = c'$, or $\varnothing$ otherwise. |
| $R_1 R_2$ | Concatenation of two regular expressions $R_1$ and $R_2$ | If the derivative of $R_1$ w.r.t. $c'$ is not $\varnothing$, then it's the concatenation of that with $R_2$. Otherwise, if $R_1$ matches $\epsilon$, then it is simply the derivative of $R_2$ w.r.t. $c'$. If not, then the result is $\varnothing$. |
| $R_1 \| R_2$ | Union of two regular expressions $R_1$ and $R_2$ | Union of the derivatives of $R_1$ and $R_2$ w.r.t. $c'$ (if one becomes $\varnothing$, simply return the other). |
| $R*$ | Klenee star - any number of repetitions of $R$ | If the derivative of $R$ w.r.t. $c$ is not $\varnothing$, then return the concatenation of that derivative with $R*$. Otherwise, return $\varnothing$. |

Table 3: Computing Brzozowski derivatives for each constructor of regular expressions under the assumption that Klenee stars are greedy. The resulting algorithm runs in linear-time on the size of the regular expression.

aliases to enforce that they will be assigned a table later. Moreover, a condition in the WHERE clause might involve a nested query, which in its turn might redefine or create new aliases. As a result, our symbol table keeps a *stack* of scopes to properly resolve aliases in nested contexts.

We constrain numeric literals to either come from a set of common numbers (including $0$ and $1$) or from the user's natural language question. This prevents the model from copying arbitrary numbers from the few-shot examples in the prompt. Finally, since SQL is case-insensitive, our CE returns case-insensitive regular expressions for keywords, table and column names.

## C.2   VEGA-LITE

Vega-Lite is a declarative language for specifying data visualizations given a data frame – a table where rows represent data points and columns represent attributes of various data types. Its syntax is a subset of JSON. Therefore, our Vega-Lite grammar accepts JSON objects that follow a subset of the Vega-Lite schema.

As in SQL, we use the user's data frame to constrain valid field names. Additionally, in Vega-Lite, one must also specify a Vega-Lite type that is used to interpret the field values. We inspect the run-time values in the data frame to determine compatible Vega-Lite types. For example, a string column is typically used as a categorical value (nominal, in Vega-Lite). However, if its entries have ISO-formatted timestamps, it can be used as temporal, or quantitative if its values can be parsed as numbers. Since JSON objects are unordered, it must handle two valid output orders: the model might output the field name first (we thus later constrain the type) or the data type first (we then constrain the field name to compatible columns). Because Vega-Lite's behavior is to silently ignore invalid data points, associating a column with an incompatible type simply produces an empty chart. Our constraint completely prevents this class of errors.

We forbid repeated fields and limit the length of free-form string literals to 30 characters, which prevents the common failure mode of language models to enter repetition loops. Similarly, we only allow an aggregation in one of the X-Y axes, but not both, since that collapses the chart to a single data point and is another common failure case. Finally, we prevent the model from faceting (i.e., splitting into multiple charts) based on a column with too many ($> 50$) distinct values: that typically crashes the Vega-Lite rendering engine since it allocates an overly large output image. In summary, our constraints guarantee conformance to the Vega-Lite specification and additionally avoid common mistakes that cause crashes or degenerate outputs.

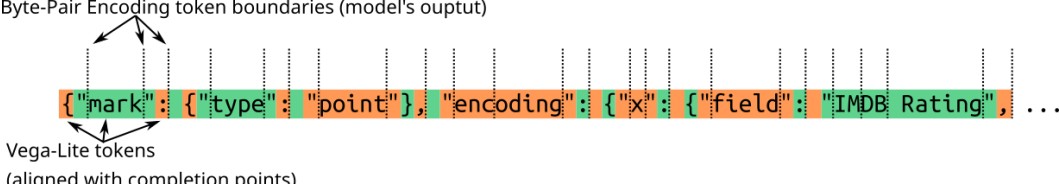

Figure 7: Illustration of the token misalignment problem in Vega-Lite. Colors denote Vega-Lite tokens, which match how the completion engine works (and what are its completion points). Vertical lines denote how GPT-3 tokenizes this program.

## C.3 SMCALFLOW

SMCalFlow programs express responses to user queries about calendar events, weather, places, and people (Andreas et al., 2020). It is a rich language with scoped variable declarations, generic types, polymorphic operators and a large API of over 400 functions, which can be composed to express complex actions like "cancel any meetings on the same day of my next doctor's appointment", or answer queries such as "will it be raining during my next walking meeting with Frank?"

Our CE enforces that programs type-check[2] by construction. An example[3] is given in Figure 1. At the current point in inference, the model is producing an argument to `size`. Since this function takes a list, its argument must be the return value of a function with return type `List<T>`. Among all functions and methods in the SMCalFlow API, only 14 return lists, which severely limits the valid options for the callee. Similarly, we keep track of declared variables inside `let` expressions, together with their types (inferred from their initialization expression), and use that data structure to limit options whenever a token of type `identifier` is syntactically allowed to follow.

Finally, we implemented heuristics based on user utterance patterns that avoid common failure cases we observed in GPT-3. These tend to happen when the model blindly copies portions of the examples in the prompt without adaptation. For instance, whenever the utterance contains exactly one of "a.m." or "p.m", this is usually represented in SMCalFlow by a call to a corresponding SMCalFlow function that constructs a `Time` object (e.g., `NumberAM(5)`). However, if the examples retrieved for the prompt tend to only have times in the opposite half of the day, GPT-3 might call the wrong function, and translate the time sub-expression in "Schedule it for 5pm" into `NumberAM(5)`. To avoid this, if we detect exactly one of "a.m." or "p.m." in the utterance, we remove the time construction functions associated with the opposite pattern from the candidates. We do the same filtering with days of the week and months, which are also constructed by specific functions.

In all domains, the CE abstraction allows us to easily encode domain knowledge in a modular fashion. Besides constraints coming from the language's semantics, it further allows domain experts to analyze failure modes of the language model and to implement fixes them in a modular and predictable manner.

## D THE TOKEN MISALIGNMENT CHALLENGE

The main challenge of CSD is in aligning constraints expressed in terms of programming language tokens with the model's output, which happens in another token vocabulary (typically constructed with Byte-Pair Encoding). Figure 7 shows an illustration of this challenge in Vega-Lite. Arbitrary mismatches can occur: the first BPE token includes the first Vega-Lite token and the first character of the second. In the middle of the example, the `"encoding"` token in Vega-Lite spans 4 BPE tokens, with misaligned boundaries at the beginning and end. Nonetheless, this issue is seamlessly handled by the CSD algorithm.

---

[2]Since the public SMCalFlow dataset does not come with the API specification, we use the type-annotated programs in the training set to infer parameter and return types.

[3]In the public dataset, SMCalFlow programs are given in a LISP-like syntax. For clarity, we mimic the original paper and use a Python-like syntax in our explanations.

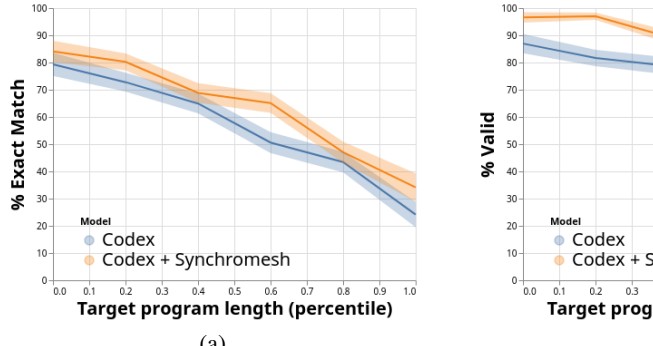

(a)                    (b)

Figure 8: (a) Accuracy and (b) validity of Codex predictions with and without SYNCHROMESH on SQL as a function of the ground-truth program length. We map program lengths to percentiles, and round to the closest multiple of 10%. Error bands correspond to standard error.

## E    OPTIMIZATIONS TO CSD

The algorithm described in Section 3.3 tests each token from the language model's vocabulary individually. However, many BPE tokens are prefixes of oen another, which lets us apply a significant optimization. If $t_1$ is a prefix of $t_2$ and $t_1$ is inadmissible after a partial program $p$, then $t_2$ is also inadmissible. Thus, we test tokens by order of length, and keep rejected tokens in a Trie structure. Before we test a token against the completion engine, we check whether one of its prefixes was already rejected. If so, we can safely skip that token.

Another optimization consists in memoizing the enumerated completion points. When testing a new partial program $p$, instead of starting from the empty string, we can start from the longest known completion point that is a prefix of $p$. This, again, can be efficiently done by keeping completion points in a Trie.

## F    CSD WITHOUT DIRECT ACCESS TO THE LANGUAGE MODEL

We used the public OpenAI API to access GPT-3 and Codex. Therefore, we did not have direct access to the underlying language models. Even so, CSD can still be applied provided we can pass a bias to be added to the logits, which is available in the OpenAI API.

However, making one request at each token is too slow. Instead, we apply a "rejection"-based sampling, as follows. First, we request a complete program from the model. Then, we iterate token-by-token, validating it with the CSD algorithm against the completion engine. If we find a violation, we (a) use CSD to determine all valid next tokens, (b) make a request asking from just a single token, applying a logit bias to constrain it to the valid tokens, and then (c) continue generation after appending the new token. Most (90%+) trajectories end after at most 3 corrections to the model. In degenerate cases, we might need to correct the model after almost every token. In our experiments, we capped CSD to apply at most 15 corrections, to control the time a request with CSD takes. This only happened in less than .5% of the cases, and could be completely avoided if we had direct access to the model (in which case CSD is efficient enough to be applied at every token).

## G    ANALYSIS OF ACCURACY AND VALIDITY BY LENGTH IN SQL

Figure 8 shows the equivalent of Figure 4 for the SQL domain. We notice that the largest gaps in validity happen for the longest queries, and the benefits in accuracy are highest for queries around the 60% length percentile.

## H  TST FINE-TUNING DETAILS

In Section 2, we described TST, which fine-tunes a sentence embedding model to attempt to capture program similarity. Here, we give more details on our training procedure.

First, for both SQL and SMCalFlow, we selected a random set of 2000 examples, and computed the normalized tree edit distance between all pairs of examples. We then fine-tuned S-BERT for one epoch using the cosine similarity loss. We used the AdamW optimizer with a learning rate of $2 \times 10^{-5}$ – the default parameters in the S-BERT library[4]. We did not perform any hyperparameter tuning. Training took around 3 hours on a single GPU.

## I  COMMENTED EXAMPLES OF TST AND CSD

Here, we provide three more detailed examples. The first examples shows the benefit of using TST on top of Codex. The remaining two show value of CSD on top of Codex. Since the examples in Sections 2 and 3 used SQL, we now give examples in SMCalFlow and Vega-Lite for diversity.

### I.1  TST EXAMPLE IN SMCALFLOW

**Query: "Hi, I need to meet with Alex on this Sunday at 6pm at the CVS shopping center for an hour."**    This query asks the system to create an event. Without TST, the top example (similar to the query) returned by vanilla S-BERT from the training set is:

```
User utterance: Hi, I need to meet with Alex tomorrow afternoon
                for an hour.
SMCalFlow: (Yield
             (CreateCommitEventWrapper
              (CreatePreflightEventWrapper
               (&
                (&
                 (Event.start_? (DateTimeConstraint (Afternoon)
                                                    (Tomorrow)))
                 (Event.duration_? (?= (toHours 1))))
                (Event.attendees_? (AttendeeListHasRecipient
                 (Execute (refer (extensionConstraint
                   (RecipientWithNameLike (EmptyStructConstraint)
                    (PersonName.apply "Alex")))))))))))
```

This example also involves creating an event, and also with a person named "Alex". However, the time structure of this example, "Tomorrow afternoon", is not trivial to translate to "this Sunday at 6pm", because these two specifications require different APIs from SMCalFlow. Given this and 4 more examples, Codex generates:

```
(Yield
 (CreateCommitEventWrapper
  (CreatePreflightEventWrapper
   (&
    (&
     (Event.start_? (DateTimeConstraint (Afternoon) (ThisSunday)))
     (Event.duration_? (?= (toHours 1))))
    (Event.location_?
     (?= (LocationWithNameLike (EmptyStructConstraint)
     (LocationName.apply "CVS shopping center"))))
    (Event.attendees_? (AttendeeListHasRecipient
     (Execute (refer (extensionConstraint
       (RecipientWithNameLike (EmptyStructConstraint)
        (PersonName.apply "Alex")))))))))))
```

---

[4]https://www.sbert.net/docs/training/overview.html

This prediction, while seeming reasonable at a surface level, violates the SMCalFlow APIs in multiple places. First, the function "ThisSunday" does not exist. The parameter 6pm was lost in this prediction, since the examples had no explicit time in it and Codex could not identify where to put it. Besides that, the model is trying to specify four attributes of the created event: start time, duration, location and attendees, but it would need one more conjunction (call to "&") to do so. This error likely came from the fact that the top example only specified three attributes. The calls to the location APIs are also incorrect – "LocationWithNameLike" exists (and is present in other examples in the prompt), but is not really needed or appropriate in this context.

However, if we use TST to pick the examples for the prompt, Codex is able to solve the problem. With TST, the top retrieved example is:

```
User utterance: Schedule 2 hours on Friday at 7:00 am for a Meeting
                with Kayla at the Sear's Tower.
SMCalFlow: (Yield
             (CreateCommitEventWrapper
              (CreatePreflightEventWrapper
               (&
                (&
                 (&
                  (Event.start_? (?= (DateAtTimeWithDefaults
                                        (NextDOW (Friday))
                                        (NumberAM 7L))))
                  (Event.duration_? (?= (toHours 2))))
                 (Event.location_?
                  (?= (LocationKeyphrase.apply "Sear\'s Tower"))))
                (Event.attendees_?
                 (AttendeeListHasRecipient
                  (Execute (refer (extensionConstraint
                   (RecipientWithNameLike (EmptyStructConstraint)
                    (PersonName.apply "Kayla")))))))))))
```

Note that the utterance, from the perspective of its natural language structure, is less similar to the user's request than the previous example. However, it describes a query with the exact necessary structure: it schedules something for the next Friday (trivial to modify to Sunday) at a certain time of the day ("NumberAM" can be simply rewritten to "NumberPM", which Codex does correctly). The location attribute, which Codex had previously tried to adapt from a less related example, is now also specified here. With this example, Codex is able to make a correct adaptation. It predicts:

```
(Yield
 (CreateCommitEventWrapper
  (CreatePreflightEventWrapper
   (&
    (&
     (&
      (Event.start_? (?= (DateAtTimeWithDefaults
                            (NextDOW (Sunday))
                            (NumberPM 6L))))
      (Event.duration_? (?= (toHours 1))))
     (Event.location_?
      (?= (LocationKeyphrase.apply "CVS shopping center"))))
    (Event.attendees_?
     (AttendeeListHasRecipient
      (Execute (refer (extensionConstraint
       (RecipientWithNameLike (EmptyStructConstraint)
        (PersonName.apply "Alex")))))))))))
```

This matches the reference program from the dataset.

## I.2 CSD EXAMPLES IN VEGA-LITE

**Query: "average horsepower over years, colored by origin"** This query does not specify a chart type, but one could infer that a line chart is likely more appropriate since it is showing a quantity over time. The dataset under consideration contains car models with their year of manufacture, horsepower, number of cylinders and many other attribute. Thus, to obtain the "average horsepower" in a given year, we need an aggregation in Vega-Lite. Codex alone predicts:

```
{"mark": {"type": "line" },
 "encoding":
   {"x":  {"field": "Year", "type": "temporal",
                            "timeUnit": "yearmonth" },
    "y":  {"field": "Horsepower", "type": "quantitative",
                                 "aggregate": "average" },
    "color":  {"field": "Origin", "type": "nominal" } } }
```

All the right components of the query are there. However, even though the query asks for the "average horsepower", the aggregation function in Vega-Lite is called "mean". Specifying an invalid aggregation function makes Vega-Lite raise an exception. Thus, an actual user wouldn't see any plot from this response.

CSD prevents Codex from generating "average" after it outputs the field name "aggregate", forcing it to choose one of the existing aggregation functions (max, min, mean, etc). Constrained by CSD, Codex generates:

```
{"mark": {"type": "line" },
 "encoding":
   {"x":  {"field": "Year", "type": "temporal",
                            "timeUnit": "yearmonth" },
    "y":  {"field": "Horsepower", "type": "quantitative",
                                 "aggregate": "mean" },
    "color":  {"field": "Origin", "type": "nominal" } } }
```

This is not only valid but also matches the ground-truth from the dataset.

**"Create a scatterplot of acceleration vs horsepower by model"** This utterance asks for a chart that relates each car's acceleration and horsepower. Codex generates:

```
{"mark": {"type": "point" },
 "encoding":
   {"x": {"field": "Horsepower", "type": "quantitative" },
    "y":  {"field": "Acceleration", "type": "quantitative" },
    "color":  {"field": "Model", "type": "nominal" } } }
```

The produced plot, shown in Figure 9 (left) is in principle correct, and answers the user's question. The "by model" in the query might sound as if the user wants specialized plots by model. Codex realizes this interpretation by coloring each point differently by the car's model. But in this dataset, each row represents a different car with a distinct model. Thus, there are 303 unique values of the "Model" column, and Vega-Lite raises a run-time warning saying that the legend has overflowed for having too many values, so some values (the majority, in this case) were omitted. When the model decides to facet on a column with too many values, the Vega-Lite run-time often runs out of memory for allocating an overly large image.

CSD avoids these errors by looking at the user's dataset and determining which columns can be used for coloring or faceting, based on how many distinct values they have. If none are available, then CSD doesn't let the model use these features. In this example, CSD forces Codex to choose one of the columns with less than 30 distinct values. Between those, Codex generates "Year". The resulting plot is shown in Figure 9 (right). While in this example the "Year" column was not mentioned in the user's request, the application of this constraint here steers Codex to avoid a run-time warning or error. The ground-truth visualization for this description does not have any color specification,

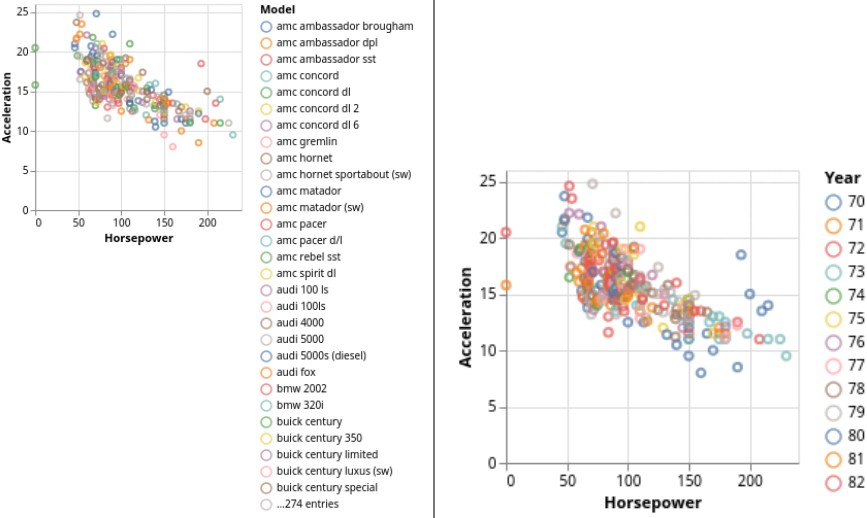

Figure 9: Benefit of CSD: The left plot is generated by Codex without CSD, but it has an overflowing legend and it also raises a warning about missing values. Using CSD, we generate the plot on right that generates no runtime warning or error.

so neither prediction counts as an exact match. Nevertheless, from the point of view of the user's experience, CSD can be helpful even when it cannot completely fix the prediction.

