# OpenReview forum: "Synchromesh: Reliable Code Generation from Pre-trained Language Models"
_ICLR.cc/2022/Conference — ICLR 2022 Poster_

### Official Review · Reviewer_vKkK · 2021-10-30

**Correctness:** 3
**Technical Novelty And Significance:** 3
**Empirical Novelty And Significance:** 3
**Recommendation:** 6
**Confidence:** 2

**Main Review:**


It is good to see a new code generation approach that considers both syntactic and semantic rules of the output language. Such an approach could prevent implementation errors in the generated programs. The proposed approach is evaluated on three languages (SQL, Vega-Lite, and SMCalFlow). The paper is generally well written.

To select examples, the authors fetch the 5 closest examples from a training set by cosine similarity. How is the training set constructed? What if there is no similar example in the training set? That is, the returned 5 closest examples all have low similarity to the question.

How many few-shot examples are selected? Are the few-shot examples the same as the 5 closest examples?

The proposed approach utilizes pre-trained models for code generation. Actually, a traditional approach (such as the one that is based on code template/pattern analysis) could also work here. Especially, considering the proposed approach also utilizes modules such as similar example retrieval and constraint analysis. Can you simply use a non-DL method to generate code based on the similar examples you retrieved, instead of using a pre-trained model? What are the benefits of using a pre-trained model here?

This paper claims to improve the reliability of pre-trained models for code generation. However, it only investigates three domain-specific languages SQL, Vega-Lite, and SMCalFlow, which all lead to relatively simple programs. It is not clear if the proposed approach can be applied to other commonly-used, general-purpose programming languages such as Java, Python, and C/C++, which all have more complex syntactical structures.

Actually, there are already some work that consider constraints in generating more complex Java and Python programs:
Lyu et al., Embedding API dependency graph for neural code generation. https://arxiv.org/abs/2103.15361, 2021.
For program synthesis, there are also some related work that are based on input and output examples:
Zaremba et al., Learning simple algorithms from examples. arXiv preprint arXiv:1511.072
Shu et al., Neural Programming by Example, Proc. AAAI 2017, Feb 2017.


**Summary Of The Paper:**

This paper proposes SYNCHROMESH, a framework for improving the reliability of pre-trained models for code generation. SYNCHROMESH first retrieves few-shot examples from a training set using Target Similarity Tuning. It then feeds the examples to a pre-trained language model and samples programs using Constrained Semantic Decoding, which can constraint the output to a set of valid programs in the target language. The authors evaluate the proposed approach by synthesizing code from natural language descriptions using GPT-3 and Codex in three real-world languages: SQL queries, Vega-Lite visualizations and SMCalFlow programs. The experimental results look promising.


**Summary Of The Review:**

Pros:

. A new approach to code generation using pre-trained models.

. Evaluated on three languages (SQL, Vega-Lite, and SMCalFlow)

Cons:

. It is unclear if the proposed approach can be applied to other general-purpose programming languages such as Java, Python.

. Some technical details are missing.

---

> ### Author Response · Authors · 2021-11-16
> **Response to Reviewer vKkK**
>
> Thank you for the great feedback!
>
> > To select examples, the authors fetch the 5 closest examples from a training set by cosine similarity. How is the training set constructed? What if there is no similar example in the training set? That is, the returned 5 closest examples all have low similarity to the question.
>
> The training set is assumed to contain examples that will also be _useful_ for answering test queries. This is a standard setting of few-shot question answering, now crucial to LLM effectiveness as discovered by Brown et al. with GPT-3. Lack of useful examples might be less of an issue in languages seen during pretraining (such as SQL), although our results showed that even in those cases helpful examples from TST significantly boost LLM’s performance. On the other hand, for languages that the LLM has seen little or not at all during training (e.g., SMCalFlow), all it can infer about the language comes from the examples, thus they become paramount.
>
> > How many few-shot examples are selected? Are the few-shot examples the same as the 5 closest examples?
>
> That's right, the 5 closest examples were taken as the few-shot examples for the prompt.
>
> > Can you simply use a non-DL method to generate code based on the similar examples you retrieved, instead of using a pre-trained model? What are the benefits of using a pre-trained model here?
>
> One can use an enumerative approach to list many valid ways to combine the retrieved input examples, but this set will be combinatorially large. Indeed, picking one prediction out of that set requires leveraging the user's natural language description. Here, the pre-trained models shine because they have been exposed to a broad range of examples of natural language. For example, in the SQL query described by "Where does the flight AA 292 land?", the model needs to infer that "where" indicates that the answer is a place, and thus the column that is most likely should be either "ArrivalCity" or "DepartureCity" -- the hint to pick the latter is given by the verb "land". Encoding these inferences without a learned model is very challenging. Some code template analyses can introduce probabilistic aspects but still underperform end-to-end learned models.
>
> > This paper claims to improve the reliability of pre-trained models for code generation. However, it only investigates three domain-specific languages SQL, Vega-Lite, and SMCalFlow, which all lead to relatively simple programs. It is not clear if the proposed approach can be applied to other commonly-used, general-purpose programming languages such as Java, Python, and C/C++, which all have more complex syntactical structures.
>
> For syntactic validity, the method we propose for deriving completion engines from grammars and then for aligning it to the language model's vocabulary, works for any ANTLR grammar out-of-the-box -- including the grammar of general-purpose programming languages like Java/Python. On top of the syntactic layer, one could also implement semantic constraints like only using declared variables.
>
> However, you're completely right that more complex constraints in these languages can require substantially more engineering to implement than in the languages that we used to demonstrate the framework. How much effort will be spent in implementing & optimizing constraints will, of course, depend on the user and the use case, but in any case substantially less than the effort to build a bespoke code generation model.
>
> > There are already some work that consider constraints in generating more complex Java and Python programs: Lyu et al., Embedding API dependency graph for neural code generation. https://arxiv.org/abs/2103.15361, 2021. For program synthesis, there are also some related work that are based on input and output examples: Zaremba et al., Learning simple algorithms from examples. arXiv preprint arXiv:1511.072 Shu et al., Neural Programming by Example, Proc. AAAI 2017, Feb 2017.
>
> Thank you for the references! While some of these papers touch on related aspects, we believe there are several fundamental differences:
> - Lyu et al. propose to embed a dependency graph for use by the decoder. This is a good idea, as it improves performance, but note that they do not constrain the decoder. Their embedder just gives it more global information about the program. It can still violate syntax and APIs during generation, since it's an unconstrained neural model.
> - Zaremba et al. directly train a neural network to execute simple algorithms like sorting and addition. Note that it does not involve explicit code generation, and that the learned neural network is not guaranteed to correctly satisfy the examples (as their experiments show).
> - Shu et al. do generate code, but the generation process is directly designed for their DSL and does not generalize nearly as universally as TST or CSD.
>
> We will add this comparison to the Related Work section in the next revision on OpenReview.

---

### Official Review · Reviewer_vLDx · 2021-11-02

**Correctness:** 3
**Technical Novelty And Significance:** 2
**Empirical Novelty And Significance:** 2
**Recommendation:** 5
**Confidence:** 4

**Main Review:**

The two contributions of the paper are quite different and have different qualities. What is nice about the paper is that it shows that the two ideas may complement each-other.

Where the paper falls short is providing understandable limitations of these ideas. I do not agree they are universally good and some of the evaluation results look confusing and difficult to parse and understand. From the evaluation, it is not clear when the two ideas are tested separately and which improvement can be attributed to the specific idea. The output model also does not reach the accuracy level of prior state-of-the-art works, often by a lot. This usually wouldn’t be an issue if the training was really few-shot, but the TST idea removes this as it fine-tunes a transformer model.

TST
===
The main idea of TST is that instead of purely textual similarity input/output examples for the few-shot learning, the fine-tuned model would provide target query similarity metrics to the input. The choice of "normalized similarity metric between programs" to be AST edit distance is not clear. Also, there seems to be something different happening than what the explanation in the paper gives. We see cases of programs with low textual similarity, but high AST similarity that the model learns to return. However, this means this model learns how to approximately synthesize programs from the inputs. The paper does not discuss what capacity/complexity this pretrained model would need, but I assume the needs may be similar to the works that do program synthesis without pretrained language models (they outperform this work, this is for similar accuracy).
minor questions: Why was AST distance chosen here. Why would it do better than pure text distance on the program text?

CSD
===
The idea to check if a sequence could potentially complete during decoding is also not new. It was already done before, for example by Karaivanov et al., Onward 2014 paper Phrase-Based Statistical Translation of Programming Languages. However, CSD here claims to enforce syntax, scope, typing rules, and contextual logic. This is an interesting claim, but this means that quite complex logic in the decoder. First, it is not clear this logic would expand to other languages with more complex syntax. Probably what the authors implement does not do precise enforcement, because this would be too complex, but instead efficiently cuts-off many search branches that would not lead to correct solutions. This is probably a nice contribution, but is not discussed in the text of the paper. It is also not clear which of syntax, scoping, type enforcement or other semantic rules contribute to the increased precision.

The main baseline to compare to here would be generate-then-check. In such a setting, one can use an unmodified parser, type checker and validator that can probably enforce more properties than the checker during decoding. Given the results shown in the paper, it is not clear why CSD would be of any significant advantage. There is also something unclear about the comparison to generate-then-check - is it compared to the full Synchromesh or only to the CSD part of it? Also, I would expect to see generate-then-check in the main evaluation table.


**Summary Of The Paper:**

The paper considers the problem of text to code translation, usually done in state-of-the-art models using a natural language input fed into a transformer model together with other similar input/output examples as a few-shot learning and code being directly output by the model.

The paper proposes two enhancements of this process.

* The first one called Target Similarity Tuning (TST) is a way to pick input/output examples similar to the input text. Previous works use a pretrained models for natural language similarity (Sentence-BERT is compared in the paper) that determine what examples to include in the few-shot prompt and the TST proposes to also fine-tune this model on code examples such that some semantic similarity in encoded - for example using the same structure of the queries [think of SQL queries to synthesize].
* The other improvement called Constrained Semantic Decoding (CSD) proposes to alter the decoder of the transformer model and to make it avoid generating programs that are impossible to complete to syntactically or semantically correct ones.


**Summary Of The Review:**

Overall, the paper gives some interesting ideas, but it does not demonstrate these ideas lead to actual wins and doesn’t discuss its limitations. Combining the two contributions TST and CSD does not lead to state-of-the-art results, while removing some of the advantages of few-shot learning. I believe this work is not generally flawed, but in this state is below the bar of ICLR.

---

> ### Author Response · Authors · 2021-11-16
> **Response to Reviewer vLDx (Part 1)**
>
> Thank you for your time, feedback, and encouraging comments! Below, we hope to further clarify the evaluation setup, the motivation behind several decisions, and the comparison to prior work. Please also check out the General Comments for background on common questions.
>
> > The choice of "normalized similarity metric between programs" to be AST edit distance is not clear. [...] Why was AST distance chosen here. Why would it do better than pure text distance on the program text?
>
> ASTs capture a more robust notion of similarity compared to using the program's text directly. First, irrelevant lexical differences (e.g., spaces between tokens, or keyword casing in SQL) are all naturally ignored by the parser. Second, using the AST focuses more on the program structure ("sketch"). For instance, if two programs have identical structure but differ in string literals (common in all 3 languages we tried), their tree edit distance is the same regardless of the value of the literals, whereas text edit distance is sensitive to the strings. We found this especially undesirable in SQL, where text edit distance sometimes would rank two very different queries as similar only because they happened to use similar literals (e.g., country names) and all the same keywords, despite more important structural differences.
>
> That said, please refer to General Comments for a broader perspective on our use of tree edit distance. In short, we don't believe this metric is a key feature of TST, but rather the general idea of attempting to capture some sufficient notion of program similarity instead of natural language alone. That similarity notion is a much simpler task than actual program generation, while still sufficient to retrieve relevant examples for the more powerful generative model.
>
> > The idea to check if a sequence could potentially complete during decoding [...] was already done before, for example by Karaivanov et al., Onward 2014 paper Phrase-Based Statistical Translation of Programming Languages.
>
> Thank you for the reference! It is true that this general idea has been explored. However, we emphasize that CSD provides a much more general framework than prior work, including Karaivanov et al. In addition to enforcing syntax (possible by Prefix Grammars as in Karaivanov et al), CSD can also enforce context-sensitive semantic constraints. For example, enforcing “only variables that have been declared can be used” fundamentally cannot be encoded with Prefix Grammars, since these are static and context-free. In contrast, CSD allows the domain expert to leverage all available machinery and metadata of the (partial) AST parser, including the symbol table of the current scope. This distinction is non-trivial and highly effective since most interesting semantic properties of programs are context-sensitive.
>
> > Which of syntax, scoping, type enforcement or other semantic rules contribute to the increased precision?
>
> We find that syntax helps the smaller model, and in particular helps GPT-3 in SMCalFlow, which it hasn't seen during training. Codex, which was trained on substantially more code, already produces syntactically valid code close to 100% of the time, so most of the improvements on top of Codex come from semantic constraints (which are not enforceable with grammar alone).
>
> Determining which constraints bring the most benefits is subtler since constraints interact with each other. Overall, none of them dominate the others, and we only included constraints that could repair a set of selected examples from the training set that we used during development.
>
> Finally, besides the specific constraints we implemented, we found that CSD gives practitioners a way to analyze common failure modes of the model and fix them without retraining. Most often it requires less than 10 lines of code (e.g., in validating contextual Vega-Lite properties).
>
> > The main baseline to compare to here would be generate-then-check. [...] Given the results shown in the paper, it is not clear why CSD would be of any significant advantage.
>
> As we show in Figure 4c, generate-then-check might need many samples to obtain a valid program, and in some cases is never able to produce one. The line for CSD with a _single sample_ would be at close to 100% for Vega-Lite and SMCalFlow here -- note that there is a large gap between that and generate-then-check with even 5 samples. We emphasize that taking 5 samples from GPT-3/Codex can take an average of 25 seconds. As we show, in many cases, even this is not enough to produce a program that the user can even execute and evaluate. In these cases, CSD significantly improves the usability of the base model.
>
> > Is [generate-then-check] compared to the full Synchromesh or only to the CSD part of it?
>
> This comparison is just to CSD -- we use the same example selection model, and directly assess the ability to produce a program that satisfies the constraints.

---

> ### Author Response · Authors · 2021-11-16
> **Response to Reviewer vLDx (Part 2)**
>
> > I would expect to see generate-then-check in the main evaluation table.
>
> We made this separation because generate-then-check takes multiple samples, as opposed to all other models in the table that take just one. This changes the meaning of the metrics. For example in Exact Match/Execution Accuracy, there is a single way to measure them with a single sample. With multiple samples, we could do a "best@K" approach, asking "out of K samples do any of them match the ground truth?", or an "average@K" approach, but in any of these cases, the K being different makes the numbers not directly comparable. In any event, “best@K” does not translate to an actionable test-time inference procedure.
>
> That said, we can make a more apples-to-apples comparison by generating samples without constraints until we have one that satisfies them, up to K samples. We'll implement and post the results we get here soon.

---

### Official Review · Reviewer_ML5m · 2021-11-02

**Correctness:** 4
**Technical Novelty And Significance:** 3
**Empirical Novelty And Significance:** 4
**Recommendation:** 8
**Confidence:** 3

**Main Review:**

To the best of my knowledge, the ideas presented by the paper are novel and interesting. Although the two sources of improvements (better example selection and constraint validation) have already been explored by prior works, the proposed methods can address some of the major limitations of prior methods intuitively: TST further incorporates the similarity of programs instead of only considering language similarity, and CSD is more efficient and effective empirically. Both methods bring significant empirical gains to the reliability of code generation. The methods are also relatively easy to implement/reproduce and drop in existing pipelines.

I have several minor concerns:
1. Does CSD also bring significant benefits to more widely used languages like python or C++? Intuitively, the language model probably has a better model of these languages due to more training data, such that CSD is not that needed or only marginally more beneficial than generate-then-test. It would be beneficial to provide extra experiments on these languages.
2. Similarly, more general-purpose languages also have more diverse ways of achieving the same effect. Then “the similarity between target programs” that TST is trying to predict seems less well defined.
3. The context-sensitive layer of CSD seems highly language-dependent and not scalable to more general cases. Yet I imagine this is perhaps more important to validness than the context-free layer. It would be interesting to see how relatively important these two layers are over the existing datasets as well as more expressive languages.


**Summary Of The Paper:**

This paper aims to improve the reliability of LLM based code generation via two aspects: select better prompting examples based on program similarity and constrain LLM to only generate valid programs incrementally. The authors provide extensive experiments and ablations to demonstrate the improvement of accuracy and validness.

**Summary Of The Review:**

Overall, the methods proposed by this paper are novel and address major drawbacks of previous methods, leading to significant improvement of code generation performance across three real-world languages. The methods are also relatively straightforward to reproduce and built upon by future works.

---

> ### Author Response · Authors · 2021-11-16
> **Response to Reviewer ML5m**
>
> Thank you for your time and insightful questions!
>
> > Does CSD also bring significant benefits to more widely used languages like Python or C++? Intuitively, the language model probably has a better model of these languages.
>
> This is an important question. Although we did not experiment with these languages, we can point to two results from related work that suggest a positive answer. First, in [1], where the authors evaluate a large language model trained on code for synthesizing short Python programs, even their largest model still produces syntactically invalid code roughly 5% of the time. Since their median program size was just 5 lines of code, this indicates that syntactic guidance alone could be helpful for generating longer programs.
>
> Further, their largest model had type or execution errors in around 25% of their tests. While it's unclear how easily fixable with CSD those errors would be, their result shows that there is still a large space of errors that simply scaling up models does not seem to fix.
>
> As a complementary result from another concurrent work, we can point to [2], where the authors tested Codex on generating Java methods. Codex still commits several simple errors, like using undeclared variables in 11.4% of the tests. This specific constraint is simple to implement in CSD, as we did for table aliases in SQL and for declared variables in SMCalFlow. Others, like preventing type errors in a more complex type system like Java's, would require more engineering. However, altogether these results show there still is significant room to improve neural code generation models by combining them with program analysis.
>
> > Similarly, more general-purpose languages also have more diverse ways of achieving the same effect. Then “the similarity between target programs” that TST is trying to predict seems less well defined.
>
> This is again a good point. See our General Comments for more detail on this issue. In short, we'd envision that TST in more general-purpose languages would benefit from using learned representations that try to be invariant to syntax. The ContraCode paper is a demonstration of this idea for JavaScript and TypeScript, where the authors use source-to-source compilers to train a model that can identify two syntactically different realizations of the same program semantics. Their positive results suggest that their learned similarity could be leveraged by TST instead of the simpler AST edit distance we used.
>
> > The context-sensitive layer of CSD seems highly language-dependent and not scalable to more general cases. Yet I imagine this is perhaps more important to validness than the context-free layer. It would be interesting to see how relatively important these two layers are over the existing datasets as well as more expressive languages.
>
> Most of the constraints we implement only depend on information generated during the construction of the partial parse tree. For example, when tracking declared variables, we only keep a stack of scope-specific symbol tables that are updated when the relevant grammar rules are triggered by the parser. This is a classic top-down symbol table derivation, as implemented in most compiler frontends with attribute grammars. Thus, we believe that simpler context-sensitive analyses would scale just as well (see result above about Codex that suggests even these still have a room for preventing errors).
>
> On the other hand, you are right that other analyses could be harder to scale. For example, in languages with type inference (including C++17 and Java 10), enforcing type checking would require running type inference algorithms on the partial program. In terms of performance, these analyses can be made fast enough: modern IDEs often run them incrementally in real-time. However, they'd require substantial engineering effort, and might not be worth it for the additional gains they might bring. Still, CSD lets users add complexity incrementally in a modular fashion, and after observing actual failure cases of the models they are using.
>
> [1] Austin, Jacob, et al. "Program synthesis with large language models." arXiv preprint arXiv:2108.07732 (2021).
>
> [2] Mukherjee, Rohan, et al. "Neural Program Generation Modulo Static Analysis." arXiv preprint arXiv:2111.01633 (2021).

---

> > ### Comment · Reviewer_ML5m · 2021-11-30
> > **Thanks for the response!**
> >
> > The response addressed my minor concerns, thanks!

---

### Official Review · Reviewer_ZrJp · 2021-11-03

**Correctness:** 4
**Technical Novelty And Significance:** 3
**Empirical Novelty And Significance:** 3
**Recommendation:** 8
**Confidence:** 4

**Main Review:**

1. Although TST works well empirically, I still cannot get the intuition behind it. The similarity score of two questions (u_i, u_j) indicates the relevance of two (p_i, u_i) in many cases. It would be great if the authors can provide more examples as the one shown in Fig 2 (I suspect the example is not that common). On the other hand, p_i and p_j could be equivalent even if their tree edit distance is large. The authors are encouraged to provide some analysis in that case.

2. The results without TST in Table 2 are reported by selecting examples via the vanilla S-BERT without fine-tuned with the TST objective  (v.s. Random selection)?

3. On the paragraph of “Example selection model”, it would be helpful if the authors elaborate more on fine-tuning S-BERT with the TST objective in Spider and SMCalFlow.

4. Results show that the proposed method still underperforms supervised methods, while both TST and supervised methods use the full training set. Given the large size of GPT-3/CodeX, it would be better if the authors further discuss the practical merits of their approach.

5. The proposed CSD method could be applied to the decoding part of supervised methods. It would be interesting to see how much gain CSD will bring to the supervised methods.

6. Maybe I missed the number in the paper, but I wonder how many constraints did you implement in CEs for the three programs? How hard was it to collect them?

7. It is a bit weird to only show integer results in Table 2 (at least keep one decimal place).

8. It’s a bit surprising to see that TST also largely helps generate valid model outputs according to Table 2. Could you please explain a bit more about it?

9. The result of GPT-3 13B + TST performs worse than GPT-3 13B  (14% v.s. 16%), which contradicts all the other results in Table 2. Why? Or a typo?

10. Based on the results in Table 2, Vaga-Lite has relatively simple program grammar. Can you know why GPT-3/CodeX still performs very low on the task? Questions are ambiguous?

11. In Algorithm 1, is the Sample() function adjusted by other advanced sampling variants (e.g., temperature sampling/top-k/top-p)? How do these sampling variants influence the performance? Are the improvements caused by the sampling variants orthogonal to CSD?




**Summary Of The Paper:**

This paper presents a framework for more reliable code generation via in-context learning of GPT-3/CodeX. The paper is motivated by the finding that GPT-3/CodeX often generate programs with syntactic and semantic errors. To resolve this problem, the authors propose 1) Target Similarity Tuning (TST) for retrieving 5 relevant examples based on program similarity and 2) Constrained Semantic Decoding (CSD) for constraining the code generation output to a set of valid programs. The authors evaluate their proposed methods on three different code synthesising tasks (Spider, Vega-Lite, and SMCalFlow). Strong complementary gains on these tasks demonstrate the effectiveness and generalizability of the proposed framework on several tasks with different target programs (SQL, Vega-lite, and SMCalFlow).


**Summary Of The Review:**

Although most of the high-level motivations and ideas of the paper have been studied in example/prompt selection and constraint-based decoding (e.g., PICAD), the authors are able to refine them and generalize them to more diverse domains and tasks by using GPT-3/CodeX without fine-tuning with a relatively small overhead to the output process.

Also, results show that the proposed method still underperforms supervised methods, while both TST and supervised methods use the full training set. Given the large size of GPT-3/CodeX, it would be better if the authors further discuss the practical merits of their approach (compared to few-shot learning with median-size language models has more meaningful practical benefits at this time if they are able to achieve competitive performance by fine-tuning <300-shot (a small number) examples.).

---

> ### Author Response · Authors · 2021-11-16
> **Response to Reviewer ZrJp (Part 1)**
>
> Thank you for the review and the great set of questions!
>
> > Although TST works well empirically, I still cannot get the intuition behind it. [...] On the other hand, p_i and p_j could be equivalent even if their tree edit distance is large. The authors are encouraged to provide some analysis in that case.
>
> Please refer to the General Comments for a better analysis of TST. In short, TST is compatible with more semantic similarity metrics, like the one learned in ContraCode. However, in our analysis, even fine-tuning based on tree edit distance was enough to make the model more robust to irrelevant changes in natural language.
>
> > The results without TST in Table 2 are reported by selecting examples via the vanilla S-BERT without fine-tuned with the TST objective (v.s. Random selection)?
>
> Correct, they are selected using vanilla S-BERT.
>
> > On the paragraph of “Example selection model”, it would be helpful if the authors elaborate more on fine-tuning S-BERT with the TST objective in Spider and SMCalFlow.
>
> Thank you, we added a better description of the training process in the new Appendix H. In particular, one important data point that we had failed to mention was that **for TST we only used a small fraction of the training set.** Since the number of pairs grows quadratically, we simply took 2000 examples from each of Spider and SMCalFlow for fine-tuning S-BERT on all pairs of similarities. (For reference, the training sets of Spider and SMCalFlow have ~7K and ~120K instances, respectively.) Please also check out the General Comments for more background.
>
> > Results show that the proposed method still underperforms supervised methods, while both TST and supervised methods use the full training set. Given the large size of GPT-3/CodeX, it would be better if the authors further discuss the practical merits of their approach.
>
> Please refer to General Comments for a broader perspective about this comparison. In short, supervised methods are used only as a reference as they use the full training set and TST does not.
>
> > The proposed CSD method could be applied to the decoding part of supervised methods. It would be interesting to see how much gain CSD will bring to the supervised methods.
>
> This is a great point. We believe the concurrent results of PICARD (just presented this week at EMNLP), which implemented some constraints for SQL, shed light on the potential of CSD even on top of supervised methods. Notably, they fine-tune T5 on the Spider dataset, achieving 71.4% execution accuracy, but implementing constraints very similar to those we described in CSD (e.g., restricting column names & keeping track of table aliases) brings it to 79.3% accuracy (in the dev set). Validity errors also disappear, from 88% to 2%. Thus, even supervised methods still make mistakes that are within the realm of constraints that CSD is designed to support.
>
> > Maybe I missed the number in the paper, but I wonder how many constraints did you implement in CEs for the three programs? How hard was it to collect them?
>
> We indeed did not count the constraints explicitly in the paper, although we listed them all in Appendix C. In total, besides syntactic validity (which we derive for free from the grammar), we implemented 5 semantic constraints for SQL, 7 for Vega-Lite and 5 for SMCalFlow. The process we followed was to use a small number of examples from the training set, run GPT-3 on them and analyze the errors it made. Most constraints were implemented in less than 10 localized lines of Python code, which mostly take the form of testing a condition on the partial AST and formatting a regular expression.
>
> > It is a bit weird to only show integer results in Table 2 (at least keep one decimal place).
>
> Thank you, this is true. We had originally kept only integers because of space (the table grew out of the margins with one more digit in each column), but we'll re-format it shortly to include one decimal place.
>
> > It’s a bit surprising to see that TST also largely helps generate valid model outputs according to Table 2. Could you please explain a bit more about it?
>
> Great question, this is indeed an important point that is worthy of more intuition. What we observed is that giving more relevant examples can drastically reduce the amount of extrapolation that the LLMs need to do. When the examples are clearly unrelated to the target query, the models try to apply many more changes, and sometimes predict pieces of the output that are not present in any of the examples. In these cases, there are more places where the model can fail. On the other hand, with TST, it is more common that the model mostly needs to compose pieces of the higher-quality examples that it receives, which the models can do more reliably than predicting new pieces from scratch.

---

> ### Author Response · Authors · 2021-11-16
> **Response to Reviewer ZrJp (Part 2)**
>
> > The result of GPT-3 13B + TST performs worse than GPT-3 13B (14% v.s. 16%), which contradicts all the other results in Table 2. Why? Or a typo?
>
> The number is correct. We observed that GPT-3 13B is more prone to copying examples without modification from the prompt when they look ostensibly relevant. We found that TST triggered this behavior slightly more often in SQL for this model. Since this model was already worse at generalizing from the examples given in the prompt, giving better examples alone (without CSD) was not sufficient to improve its performance in SQL. This was not true in other languages (in particular in SMCalFlow, which it hadn't seen during training and thus relied completely on relevant examples).
>
> > Based on the results in Table 2, Vaga-Lite has relatively simple program grammar. Can you know why GPT-3/CodeX still performs very low on the task? Questions are ambiguous?
>
> There are a few factors at play. One is, as you guessed, that questions are ambiguous, but we're only considering exact match accuracy. Many descriptions in the NLV Corpus are formulated as questions, such as _"Is there a correlation between IMDB Rating and Rotten Tomatoes rating?"._ The ground-truth plot answers this question, but so could other visualizations that aren't exactly identical. Thus, Exact Match accuracy should be interpreted as a lower bound on "Semantic Accuracy", which would ideally be measured with a human evaluation.
>
> Aside from that, note that even though Vega-Lite's syntax is simple (a subset of JSON), its semantics aren't trivial. Many JSON fields interact in subtle ways that can make the output unusable. For example, it is valid to add an aggregation, like computing the mean, on the X axis, as it is on the Y axis. However, models sometimes add it to both axes, which causes the plot to collapse. GPT-3 makes many such errors. Codex is better but still generates broken code in ~13% of cases. CSD nearly eliminates these classes of errors.
>
> > In Algorithm 1, is the Sample() function adjusted by other advanced sampling variants (e.g., temperature sampling/top-k/top-p)? How do these sampling variants influence the performance? Are the improvements caused by the sampling variants orthogonal to CSD?
>
> Those are all valid options. In our experiments, for Table 2, we used a temperature of 0 to just get the highest-probability sample (done by passing a temperature of 0 to the OpenAI API), and for Figure 4c we used a temperature of 0.7. We did not modify top-k or top-p. In early experiments, temperatures lower than 0.5 tended to yield mostly duplicate samples (which was undesirable when the model was confident yet wrong), and a temperature of 0.9 or 1.0 tended to degenerate more often (when the model had lower confidence, samples had high variance). We used a middle ground of 0.7, but nucleus sampling could also be explored.

---

> > ### Comment · Reviewer_ZrJp · 2021-11-30
> > **Thanks for the responses.**
> >
> > Thanks for the detailed responses! They resolved most of my questions.

---

### Public Comment · ~Torsten_Scholak1 · 2021-11-13
**Comment from The Author of PICARD**

Like PICARD (Scholak et al., 2021, https://arxiv.org/abs/2109.05093), SYNCHROMESH aims to address the problem of valid code generation from large pre-trained language models. The approaches of PICARD and Constrained Semantic Decoding (CSD) in this paper are very similar. Both are based on the idea of constraining decoding by means of a grammar and a parser, and both can be applied to different target languages. Unfortunately, the authors of SYNCHROMESH make an ill-advised attempt to differentiate the two approaches. In section 5, "Related Work", it is claimed that, "to alleviate the token alignment problem and improve performance, [Scholak et al.] fine-tune the underlying LM." This is not the case:

First, what the authors refer to as the "token alignment problem" -- the fact that the model token boundaries and the lexical token boundaries in the target grammar do not line up -- is solved in a trivial and transparent way by the PICARD method. The PICARD checker uses the LM's tokenizer to detokenize the output of the LM incrementally, that is, token by token during generation. Because of that, the incremental parser used by PICARD can work with the detokenized subword-token sequences as is, and, therefore, additional fine-tuning of the LM to address the "token alignment problem" is not necessary.

Second, the authors further claim that, in contrast to PICARD and as a generalization thereof, "SYNCHROMESH does not require fine-tuning the underlying LM on the target domain." PICARD does not require fine-tuning of the LM either. PICARD is a model-agnostic, rule-based constrained decoding method. It can work with any LM that can generate code. However, Scholak et al. used a model, T5, that cannot generate code without fine-tuning. This is the only reason why fine-tuning is required in Scholak et al.'s work (indeed, without finetuning, T5 would try to translate the input to German, which is outside of the SQL target domain). Therefore, a charitable read of the authors' claim is that PICARD would not work well if the LM predicts the highest probabilities for many incorrect tokens. However, it is not clear to me whether or not that would be an issue for SYNCHROMESH as well. A model that does not have a good command of the output language is not a good model for code generation, with or without PICARD or SYNCHROMESH.

It would be great if the authors could correct these misrepresentations in their work.

Finally, I would like to request a change in Table 2 of the paper. In the fourth row, for the model "Scholak et al. (2021)", the authors used the test set execution accuracy of the T5-3B + PICARD model. For consistency, I think the development set accuracy should be used instead. It is 79.3 percent. Furthermore, the percentage of valid SQL programs generated by T5-3B + PICARD is 98 (up from 88 percent without PICARD). This number is absent from the table and should be added for completeness. Note that all these numbers have been reported in the published PICARD paper.

Overall, SYNCHROMESH is an interesting paper, and its CSD implementation and evaluation are different from PICARD. I am looking forward to the release of its code, and perhaps we can come up with an even better framework for code generation that combines the best parts of both approaches.

---

> ### Author Response · Authors · 2021-11-16
> **Clarifying PICARD and Synchromesh differences**
>
> Thank you for your comment, Torsten! And first of all, please accept our best regards on PICARD – it is without doubt a great concurrent work that explores a similar topic. Indeed, we agree that approaches of PICARD and Synchromesh together can lead to a general framework for code LM validity. In our textual comparison, we never aimed to misrepresent PICARD, so hopefully the answers below can clarify their respective similarities and differences. Please correct us if we misunderstood any details.
>
> ### On token alignment differences between PICARD and Synchromesh
>
> Both PICARD and Synchromesh incrementally detokenize BPE tokens from underlying LMs to obtain a program prefix to be parsed into a partial AST. PICARD uses an incremental monadic parser (attoparsec), whereas Synchromesh uses a robust non-incremental parser (ANTLR). Notably, the implementation of constraints aka “guards” in PICARD is functionally embedded into the language grammar via the same parser-combinator paradigm. This makes incremental constraint validation as seamless as parsing as a whole -- the parser will reject a violation as soon as it detects it according to the grammar rules but no sooner (even if a subtoken could be interpreted as a false positive).
>
> However, this choice fuses parsing and validation into one monolithic grammar. In contrast, a core motivation of Synchromesh is a _language-agnostic framework_ that would allow domain experts (i.e., compiler/IDE/app engineers) to define constraints **(a)** while reusing as much of the established compilation infrastructure as possible, **(b)** with no NLP knowledge, and **(c)** minimal engineering effort. The CSD framework of Synchromesh handles partial parsing for the engineer (using predefined ANTLR grammars), exposing only the APIs for definition of constraints from a partial AST. We consider a partial AST with common metadata (such as symbol tables) a universal program representation for engineers. For such separation, a partial AST must be materialized and reliable -- which requires Synchromesh to handle the detection of completion points. PICARD instead trades off some of that encapsulation for the elegance of the core implementation. The practical effect of this is exacerbated with more complex grammars (SQL → SMCalFlow → Python → C++).
>
> We will correct our misunderstanding of the motivation and handling of tokenization in PICARD, and will update the comparison to this more accurate one in the next OpenReview revision. As the preprint and source code of PICARD were published only a few days before the ICLR submission deadline, we unfortunately misread some of these motivations.
>
> ### On the use of model supervision
>
> We agree that PICARD as defined -- a constraint-driven technique for incremental beam filtering -- does not make any assumptions about the supervision of its underlying semantic parsing model. But, respectfully, its effectiveness on top of a non-finetuned model is unknown as it was never measured in the EMNLP paper. We can speculate about performance of either PICARD or Synchromesh in such a setting (and we broadly agree with your conjecture!), but in the end, it requires empirical evaluation.
>
> Synchromesh implements domain adaptation of a pretrained model with few-shot prompting instead of fine-tuning. As such, TST -- technically completely independent from CSD -- is a crucial component of Synchromesh as a whole. However, TST requires significantly less computational resources than fine-tuning T5. The resulting Codex+Synchromesh still underperforms PICARD, but note that it only completes one trajectory via temperature sampling in contrast to PICARD’s top-4 beam search.
>
> Finally, all these observations only accentuate the complementary nature of PICARD and Synchromesh approaches, in regards to either performance, adaptation cost, or engineering effort. We are looking forward to a potential collaboration on their combination.
>
> ### On Table 2
>
> We will update reference numbers, thank you for bringing this to our attention. All supervised results were present for reference only, thus do not substantially change any conclusions in the paper.

---

> > ### Public Comment · ~Torsten_Scholak1 · 2021-11-18
> > **Thank you!**
> >
> > Thank you very much for your thorough and thoughtful reply! I am looking forward to a potential collaboration as well.

---

### Public Comment · ~Uri_Alon1 · 2021-11-15
**Questions**

Thank you for this paper!

The paper presents elegant approaches for improving the generation of programs from large language models.
I especially liked the generality of the proposed TST+CSD approaches, which seem to be beneficial for many future models because they are not tailored to any specific model or language.

I was left with three questions:

1. Does Synchromesh especially fit domain-specific languages such as SQL, Vega-Lite and SMCalFlow? This is not a negative thing, I am just trying to understand if this choice of languages is a practical implementation choice, or does Synchromesh works especially well for languages in which the LLM (e.g., codex) is particularly weak? or languages with some syntactic properties? What do the authors think about this?

2. The conclusion section concludes that "We envision extending SYNCHROMESH to a Turing-complete language like Python ... The TST technique, however, can be used in any LLM-based language-to-code system". What are the current limitations that prevent the extension of TST and CSD to other languages such as python? Just a more difficult implementation and more edge cases to handle, or are there other any inherent limitations?

3. I would also be very interested to see more examples of predictions that Synchromesh had made, along with the retrieved TST prompt (as in Figure 1), and if possible, the predictions with and without TST and CSD. I think that this will allow the readers to get more intuition about the way that each of TST and CSD help.

Thanks and good luck!

Uri

---

> ### Author Response · Authors · 2021-11-16
> **On general-purpose languages**
>
> Thanks for the comment, Uri! We’ll address your questions 1 and 2 right now as they also align well with some of our answers to other reviewers. Please also check out our responses to reviewers ML5m and vKkK as they touch upon relevant questions. And we will come back with representative examples to question 3 shortly in a couple days -- incorporating them into the next OpenReview revision.
>
> At a high level:
>
> * TST requires **(a)** a small training set of representative question-program pairs for tuning a similarity metric, and **(b)** a pool of such pairs for few-shot example selection at inference time. These are very general requirements, common to all few-show NL→Code systems. As such, TST is already broadly applicable to any language.
>
>   Its effectiveness might vary due to weaker alignment of different syntax with different semantics in general-purpose languages (as reviewer ML5m points out). We have not measured the degree of its effectiveness on Python or C++ but we still conjecture that it will significantly improve example selection in many cases -- especially if fine-tuned with a semantically inspired metric like ContraCode instead of tree edit distance.
>
> * CSD requires **(a)** a robust parser that produces _partial ASTs_, **(b)** parser API to generate _follow sets_ of terminals or productions for any program prefix, **(c)** user-defined context-sensitive completion engines, parameterized by ASTs and follow sets. Out of these, completion engines (c) are actually the easiest -- given ASTs and follow sets, most interesting semantic constraints can be implemented with less than 10 lines of simple code.
>
>   The ANTLR infrastructure automatically provides (a) and (b), but its community-defined language grammars are not official and are not necessarily up-to-date with every language specification. The official compilers, in contrast, are usually up-to-date and robust (because they must handle incomplete programs in an IDE) but we could not find an official compiler for a general-purpose language that already exposes universal follow-set functionality. Many compilers only expose “completion” functionality for single tokens in selected locations, which would allow us to implement only a very weak version of CSD. Extending an official compiler with a universal follow-set API is certainly possible, and would require perhaps 1-2 person-months of work for an experienced compiler engineer. We chose to focus our engineering effort for this publication on a diverse selection of smaller yet still practical languages.
>
>   While LLMs certainly make fewer errors with languages they have seen in pre-training, they still make plenty of them. We point out two concurrent studies (on Python and Java) in our response to reviewer ML5m. Some errors are syntactic and would be fixed even with a context-agnostic CSD. The majority are semantic, and could be fixed by leveraging parsing metadata (e.g. attributes and symbol tables) in context-sensitive CSD constraints. We believe both will yield substantial benefits for code generation, as long as the one-time cost of (a)+(b) infrastructure is paid.

---

### Author Response · Authors · 2021-11-16
**General Comments**

We thank all reviewers for the detailed and thoughtful comments about our work. We appreciate the numerous independent values that multiple reviewers found in our contributions, including methods that "are novel and address major drawbacks of previous methods" [Reviewer ML5m], promising experimental results at a fraction of training cost, and general complementary applicability of TST and CSD in existing code generation pipelines. In this comment, we address some common questions, in addition to individual responses below.

### TST and Tree Edit Distance

The key idea of TST is that a simple model can learn to _measure how likely two NL questions are to be answered by similar programs._ This comparison task is easier than the generative task of NL→Code because the similarity model does not need to understand the NL intent perfectly. As such, it can be used to source few-shot examples for the LLM, which (by virtue of being a large pretrained model) is better positioned to solve the generative task from those examples. As long as examples are likely to contain _constituents_ of the target program, the LLM can often compose them into the right answer.

In TST, we used a simple tree edit distance metric to measure program similarity. Its main benefit is that it can be defined uniformly across all languages. However, its downside is that it can fail to capture _semantic_ similarity in many cases. As outlined above, this is still sufficient to capture some desired notion of relevance to the target program, and provides much more information than similarity of natural language alone. In principle, tree edit distance could be replaced with more sophisticated code-specialized distance metrics, especially in general-purpose languages. For instance, the recently presented ContraCode [1] uses a contrastive learning objective and can learn useful semantic representations for JavaScript and TypeScript. Naturally, such metrics would only improve the effectiveness of TST – of course to the extent that it is bounded by effectiveness of the generative LLM.

Even then, we observed that fine-tuning on tree edit distance made the similarity metric already much more robust to irrelevant changes in natural language. For instance, one description of a Vega-Lite visualization in NLV is _"scatter plot sales vs profit"._ Vanilla S-BERT retrieves 5 very diverse examples for this query (two histograms, two scatter plots, one line plot). If we instead request _"scatter plot **of** sales vs profit",_ suddenly all 5 retrieved examples have the word "of", and 4 of them are histograms (whose description starts with "histogram of ..."). If we now join two words into _"scatterplot of sales vs profit"_ [sic], all 5 retrieved examples contain the word "scatterplot". This is perhaps unsurprising, since S-BERT is trained on a paraphrase detection dataset, and concise program descriptions are out-of-distribution for it. In contrast, S-BERT fine-tuned on _program similary_ is robust to irrelevant NL variations in our experiments.

Finally, note that the similarity model is not in any way related to the LLM – we use a 300M S-BERT as a similarity model and up to 175B GPT-3/Codex as a generative LLM.

### Comparison to Supervised Methods

Neither LLMs nor LLMs augmented with Synchromesh reach the performance of supervised semantic parsing models. Our goal is not to achieve state-of-the-art accuracy with a single model; our goal is to **(a)** significantly improve accuracy with minimal computational resources and engineering effort, **(b)** significantly improve reliability, trust, and validity of the code generated by LLMs even when it’s not 100% correct. Both are crucial practical issues in real-world deployments of LLM applications. As such, we include supervised models’ results in Table 2 for reference, but we read them as upper bounds / targets.

Assuming LLM as an underlying code generation model eliminates most of the machine learning and architectural cost of the NL→Code application (in contrast to bespoke models). The Synchromesh framework adds a minor overhead over it – namely, the cost of fine-tuning an off-the-shelf similarity model on a few examples in TST and the cost of implementing domain-specific compiler constraints in CSD.
Unfortunately, we have failed to mention an important detail in the original submission: **for TST we only used a small fraction of the training set.** Since the number of pairs grows quadratically, we simply took 2000 examples from each of Spider and SMCalFlow for fine-tuning S-BERT on all pairs of similarities. (For reference, the supervised training sets of Spider and SMCalFlow have ~7K and ~120K instances, respectively.) We will update the training details in the new Appendix H in our next OpenReview revision.

[1] Jain, Paras, et al. "Contrastive code representation learning." EMNLP 2021.

---

### Decision · Program_Chairs · 2022-01-20

**Decision:**

Accept (Poster)

**Comment:**

The paper gives a new method for code generation from natural language queries using pretrained models. The approach follows two steps: (1) given a query, it selects a set of similar training examples using a method called Target Similarity Tuning, and (2) it then uses a method called Constrained Semantic Decoding (built on top a frozen language model) to adapt these examples into syntactically/semantically correct code.

The reviewers found the paper interesting. There were some concerns about the method's scope and its relationship to prior work but these were mostly addressed during the author response period. Given this, I am delighted to recommend acceptance. Please incorporate the feedback in the reviews (in particular, the review by vLDx) in the final version of the paper.